# Spatially coordinated dynamic gene transcription in living pituitary tissue

Karen Featherstone[1], Kirsty Hey[2], Hiroshi Momiji[3], Anne V McNamara[4], Amanda L Patist[1], Joanna Woodburn[1], David G Spiller[4], Helen C Christian[5], Alan S McNeilly[6], John J Mullins[7], Bärbel F Finkenstädt[2], David A Rand[3]*, Michael RH White[4]*, Julian RE Davis[1]*

[1]Centre for Endocrinology and Diabetes, University of Manchester, Manchester, United Kingdom; [2]Department of Statistics, University of Warwick, Coventry, United Kingdom; [3]Warwick Systems Biology, University of Warwick, Coventry, United Kingdom; [4]Systems Biology Centre, University of Manchester, Manchester, United Kingdom; [5]Department of Physiology, Anatomy and Genetics, University of Oxford, Oxford, United Kingdom; [6]MRC Centre for Reproductive Health, Queen's Medical Research Institute, University of Edinburgh, Edinburgh, United Kingdom; [7]The Molecular Physiology Group, Centre for Cardiovascular Science, Queen's Medical Research Institute, University of Edinburgh, Edinburgh, United Kingdom

**Abstract** Transcription at individual genes in single cells is often pulsatile and stochastic. A key question emerges regarding how this behaviour contributes to tissue phenotype, but it has been a challenge to quantitatively analyse this in living cells over time, as opposed to studying snap-shots of gene expression state. We have used imaging of reporter gene expression to track transcription in living pituitary tissue. We integrated live-cell imaging data with statistical modelling for quantitative real-time estimation of the timing of switching between transcriptional states across a whole tissue. Multiple levels of transcription rate were identified, indicating that gene expression is not a simple binary 'on-off' process. Immature tissue displayed shorter durations of high-expressing states than the adult. In adult pituitary tissue, direct cell contacts involving gap junctions allowed local spatial coordination of prolactin gene expression. Our findings identify how heterogeneous transcriptional dynamics of single cells may contribute to overall tissue behaviour.

*For correspondence: d.a.rand@ warwick.ac.uk (DAR); mike.white@ manchester.ac.uk (MRHW); Julian. Davis@manchester.ac.uk (JRED)

**Competing interests:** The authors declare that no competing interests exist.

## Introduction

Gene expression in single living cells is often pulsatile and heterogeneous between cells (*Sanchez and Golding, 2013*; *Coulon et al., 2013*). In a complex three-dimensional (3D) tissue, dynamic and heterogeneous single cell behaviour gives rise to the overall tissue-level gene expression state and determines the ability of the tissue to respond appropriately to acute and chronic stimuli. Transcriptional bursting, defined by periods of RNA synthesis followed by usually longer silent periods, occurs at many genes with characteristic gene-specific timing (*Suter et al., 2011*). These dynamics have been proposed to be influenced by intrinsic factors that appear stochastic and extrinsic factors that reflect the state of the cell. Thus far, the key to identifying these processes has been single cell analysis (*Raj and van Oudenaarden, 2009*; *Spiller et al., 2010*). In situ hybridisation techniques have revealed non-equivalent activity at gene alleles within individual cells (*Wijgerde et al., 1995*; *Raj et al., 2006*), but only provide snap-shot measurements of activity. The analysis of gene expression in single living cells using real-time direct RNA imaging systems confirms these pulsatile kinetics (*Chubb et al., 2006*; *Larson et al., 2013*; *Martin et al., 2013*). However,

**eLife digest** Although humans have thousands of genes, only a fraction of these are expressed in any given cell. Each cell type expresses only the genes that are relevant to its particular job or that are necessary for general cell maintenance. Even these genes are not expressed all the time: most cells express genes in bursts, and the cells that make up a tissue produce these bursts at different times. This makes it easier for the tissue to respond to new conditions.

The pituitary gland, found at the base of the brain, is often studied to investigate changes in gene expression. The pituitary gland is found in all animals that have a backbone, and it makes and releases many different hormones. For example, one type of pituitary cell expresses the gene that encodes a hormone called prolactin. This hormone has a range of roles, including stimulating milk production and regulating fertility in mammals. The coordinated production of prolactin by pituitary cells is important for reproduction, but it is not clear how (or whether) individual prolactin-producing cells in the gland communicate to coordinate bursting patterns of expression of the prolactin gene.

Featherstone et al. marked the prolactin-encoding gene in the pituitary cells of rats with a gene that encodes a fluorescent protein; this enabled the gene's activity to be observed in thin slices of living tissue using a microscope. Mathematical models were then used to analyse the recorded expression patterns.

The results showed that in a single cell, the bursts of expression of the prolactin gene are randomly timed. This means that although the expression activity of an individual cell is unpredictable, the overall activity of a group of cells can be precisely determined. The model also showed that cells coordinate when they express the prolactin gene to a greater extent with their near neighbours than with cells that are further away in the tissue. Featherstone et al. found that this coordination depends on structures (called gap junctions) that connect the cells and allow signalling between them, and this tissue organisation is established during early development.

The mechanisms underlying the timing of the bursts remain to be discovered. The timing for the prolactin gene seems to be dominated by a minimum delay that must occur before the next burst can be reactivated. Future challenges also include determining whether coordinated gene expression occurs in other tissues and whether this coordination is disrupted in disease.

direct RNA analysis is technically challenging and relatively low throughput, even over short time periods. An alternative is the use of reporter gene analysis. Whilst being indirect, this can be combined with mathematical modelling (*Suter et al., 2011*; *Harper et al., 2011*) to give a quantitative description of the dynamics of single cell gene expression.

The pituitary gland is an excellent model system in which to address how dynamic changes in gene activity are regulated in vivo. The gland is composed of multiple cell lineages that are regulated by external signalling inputs from the hypothalamus and circulation (*Featherstone et al., 2012*), as well as through complex paracrine signalling within the gland (*Denef, 2008*). The spatial positioning of cells is organised with cell networks facilitating the propagation of signals across the gland (*Le Tissier et al., 2012*; *Mollard et al., 2012*). The distinct cell types of the pituitary secrete specific hormones in an organised manner in response to developmental and environmental cues, which has proved to be a useful model system for investigating pituitary-tissue-specific and regulated gene transcription dynamics (*Featherstone et al., 2012*). Prolactin (PRL) is an important pituitary-derived hormone with multiple functions that is secreted from pituitary lactotrophic cells and controlled in a complex way in response to both acute and long-term signals (*Featherstone et al., 2012*; *Ben-Jonathan et al., 2008*). The human PRL (hPRL) gene displays bursting activity in cell lines and primary cells, with variable periods of active and inactive transcriptional states, including a refractory period in the inactive state (*Harper et al., 2011*). For hPRL, the dynamics observed in dispersed cells are compatible with a binary mathematical model in which there is an 'off' and an 'on' state together with a preparatory or 'primed' state. The transcription machinery transitions between these states as off – primed – on – off, with the time in each state exponentially distributed (*Suter et al., 2011*; *Harper et al., 2011*). We call this the telegraph process with priming.

Quantitative imaging of hPRL reporter gene expression (*Spiller et al., 2010*) has been used to describe PRL gene activity in different physiological states of the pituitary gland (*Featherstone et al., 2011*; *Harper et al., 2010*). Here, we employ a new mathematical and statistical model to estimate not only the timing but also different levels of transcriptional activity. This provides a quantitative framework by which to explore temporal and spatial PRL gene expression in single cells within the anterior pituitary gland. We show that shorter durations of activity occur at high transcription rates in immature pituitary glands compared to the adult pituitary; however, we were unable to detect differences in the distribution of transcription rates in different pituitary states. These results suggest that dynamics of gene activity may have a common mechanistic basis in tissues and single cells at all stages of development, which are directly regulated by the developmental state of the tissue. Moreover, these dynamics demonstrate that transcription does not occur as a simple binary 'on-off' process with a single transcription rate in the 'on' state.

In this study we also investigated how the spatial organisation of lactotroph cells within the pituitary affect the transcription dynamics of the hPRL gene. Transcription activity was coordinated between closely localised cells within the adult gland, but in developing pituitaries this transcriptional coordination was not evident, potentially due to an immature network of cellular communication. Perturbation of cell communication in the adult gland through trypsin-mediated digestion of extracellular proteins or pharmacological inhibition of intercellular gap junctions reduced transcriptional coordination between cells. These studies provide insight into how transcription dynamics throughout development and adult life relate to the state and structure of a complex and physiologically important tissue.

## Results

### Patterns of prolactin gene transcription activity in adult pituitary tissue

To quantify PRL gene transcription dynamics within living pituitary tissue, we used transgenic rats that contain a destabilised EGFP (d2EGFP) reporter gene expressed under the control of the hPRL gene locus (hPRL-d2EGFP) (*Semprini et al., 2009*). Imaging analyses were initially performed on adult male pituitary slices (300 μm coronal sections) (as depicted in *Figure 1A*) for up to 48 hr in basal culture medium, with fluorescence measured from all actively transcribing cells within a field of approximately 100 cells. Tissue slices were subsequently stimulated with forskolin (5 μM), inducing at least a twofold increase in signal, showing that tissue slices remain viable in culture for long periods (*Figure 1B*). Cells showed differing profiles of fluorescence activity during the imaging period (*Figure 1C*). Generally cells increased hPRL-d2EGFP reporter gene expression, with the mean fluorescence activity reaching maximal levels at approximately 24 hr, potentially due to the removal of the pituitary from the inhibitory effects of hypothalamic dopamine (*Harper et al., 2010*). Autocorrelation analyses did not identify a regular homogeneous period, and fluorescence activity showed a clear deviation from a white noise process (*Figure 1D*). Further quantitative analysis using a stochastic switch model (described in [*Hey et al., 2015*]) uncovered a clear statistical structure indicating a pulsatile transcriptional behaviour compatible with a pulsed telegraph process. The mathematical analysis of this is addressed in the modelling studies described below. Overall, we found that PRL transcription is dynamic within individual cells, with heterogeneous activity across the cell population.

### Characterisation of prolactin gene transcription dynamics in adult pituitary glands

The measurement of fluorescent protein activity by confocal microscopy enables the quantification of real-time gene transcription dynamics through mathematical modelling. The original transcription rate of the reporter gene is determined by considering the contribution of mRNA translation rates and mRNA and protein degradation rates to the observed fluorescent protein activity. Binary modelling of PRL transcription dynamics using this approach identified important parameters of gene activity, including a refractory period in the 'off' state (*Harper et al., 2011*). We have extended this approach through the development of a stochastic switch model (*Hey et al., 2015*), which enables the inference of different levels of transcription rate as well as the timing of switches between different levels of activity. The model uses a reversible jump Markov chain Monte Carlo (*Green, 1995*)

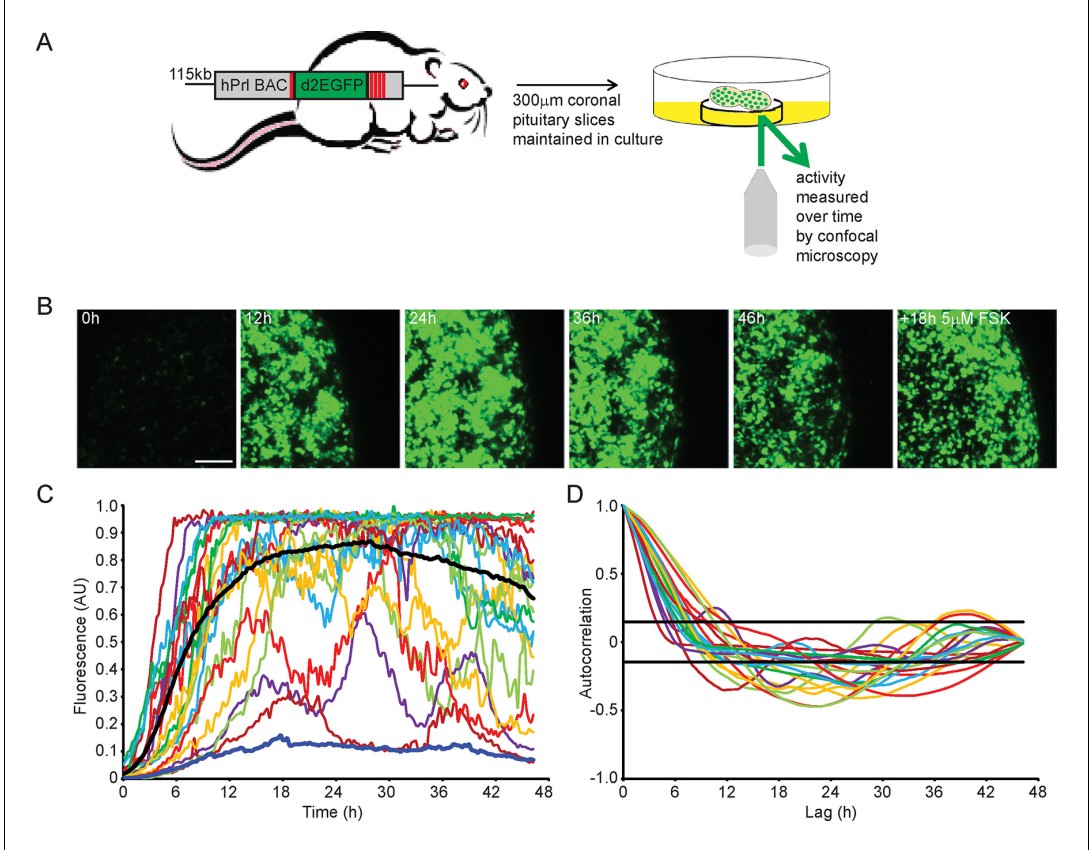

**Figure 1.** Patterns of prolactin gene transcription activity in pituitary tissue. (**A**) Schematic of the reporter construct and experimental approach used. (**B**) Images of d2EGFP expression in lactotroph cells of adult male pituitary tissue during a 46-hr imaging period, in basal culture media, and a subsequent 18-hr imaging period following stimulation with forskolin (FSK). Images shown are a maximum intensity projection of a z-stack over 242 µm. Bar represents 100 µm. (**C**) Graph of d2EGFP fluorescence (average intensity, arbitrary units) from 20 individual cells (representative of 101 cells analysed, from one experiment) from adult male pituitary tissue. The black line represents the mean activity from all the cells analysed. The dark blue line represents the background fluorescence intensity (mean from five areas). (**D**) Autocorrelation analysis of d2EGFP fluorescence intensity from cells shown in (**C**), with the 95% confidence interval representing a white noise process shown between black lines. Data are representative of three independent experiments. For validation of single cell imaging see *Figure 1—figure supplement 1*. d2EGFP, destabilised EGFP.

The following figure supplement is available for figure 1:

**Figure supplement 1.** Validation of single cell imaging.

algorithm to produce a probability distribution over all possible transcriptional profiles for each cell. Post-processing of this distribution enables the extraction of a number of candidate transcriptional profiles, each with an associated probability of occurrence (as depicted in *Figure 2A*). Consequently, the transcriptional analysis for each cell is a weighted analysis of all possible transcriptional profiles taking into account the probability of occurrence of each profile (further details of the processing of the data and algorithm output are given in 'Materials and methods'). The fit of the model was tested through calculation of recursive residuals as a way of comparing the prediction from the model and the observed data (for more information see Appendix G of [*Hey et al., 2015*]). These showed no departure from the model assumptions, indicating that the stochastic switch model fitted the data well.

We applied the stochastic switch model to the time-lapse fluorescence imaging data to understand the switches in hPRL gene expression in the intact tissue and investigate whether multiple levels of transcriptional rate were employed during phases of active transcription. The model also estimates the timings of switches between statistically different transcriptional states. The fluorescence data were processed to ensure that they were linear and quantitative as described in

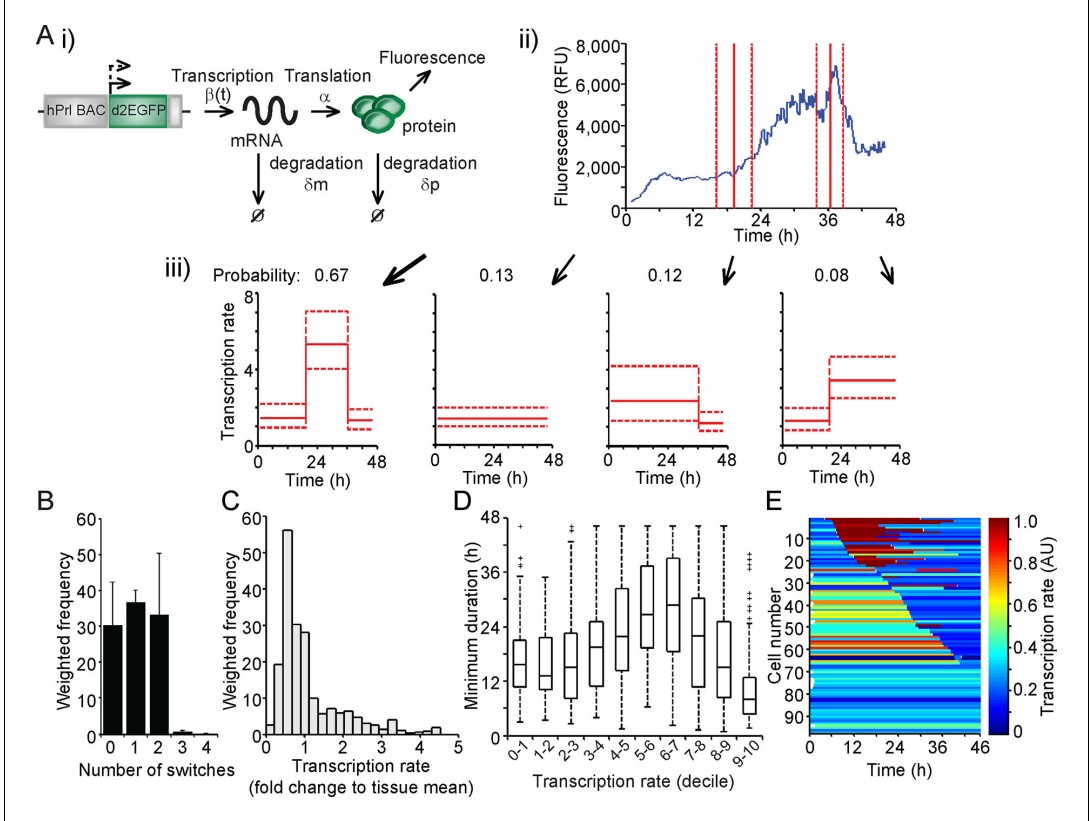

**Figure 2.** Characterisation of prolactin transcription dynamics. (**A**) i) Schematic of the parameters (translation rate, mRNA and protein degradation rates) used by the stochastic switch model to back-calculate to the original transcription rate, β(t), of the reporter construct. For each fluorescence time series, there exist several mutually exclusive possible transcriptional profiles based on the sampled switch times. ii) The graph shows the relationship between the measured fluorescence activity (blue) and possible switch times (red, dashed lines represent ± 1.96 SD). iii) The four possible profiles associated with the estimation of two transcriptional switches are shown along with their predicted probability of occurrence. (**B–E**) Characterisation of transcription rate activity of the reporter construct in cells maintained within adult pituitary tissue. (**B**) The number of switches in transcriptional states estimated during the imaging time-course. The frequency of switches is weighted by the probability of occurrence of each profile. Data are represented as the mean + SD from three independent experiments. (**C**) Distribution of the frequency of transcription rates estimated from fluorescence activity, with a weighted density calculated from all possible transcriptional profiles for each cell. The cumulative distribution of transcription rates from three independent experiments is shown in *Figure 7B*. (**D**) The duration of transcriptional states, with the caveat that the duration is the minimum duration as complete periods of activity were not detectable for most states. Transcription rates were binned into deciles and the associated minimum durations calculated. Boxplots are calculated by random sampling of the weighted transcription rate distributions for each cell. There is some evidence for longer durations at mid-range transcriptional rates (4th–8th decile), although longer observation windows may yield further trends. Data shown were pooled from three independent experiments. Boxplots represent the median and interquartile range (IQR), with whiskers drawn 1.5xIQR away from the lower and upper quartile. (**E**) Heatmap of cell transcription rate patterns ordered by the timing of the first switch event. A single transcriptional profile was selected at random for each cell taking into account the probability of occurrence. Data shown in (**C**) and (**E**) are from a single experiment on adult pituitary, representative of three independent experiments. SD, standard deviation.

The following figure supplement is available for figure 2:

**Figure supplement 1.** Processing of fluorescence time-series prior to transcription rate estimation.

*Figure 2—figure supplement 1*. In adult pituitary tissue, we found that the majority of cells switch between different levels of PRL gene transcription rate up to two times within the 46-hr imaging period (*Figure 2B*), consistent with the autocorrelation analysis. The transcription rates displayed by individual cells showed a continuous range of activity and were not restricted to simple binary on-off behaviour (*Figure 2C*). Durations of activity varied with different transcription rates (*Figure 2D*), and switches in activity were heterogeneous in both their timing and amplitude (*Figure 2E*).

## Spatial organisation of prolactin transcription activity in adult pituitary

A key aim was to establish the extent and mechanism of cell communication in the control of PRL gene expression in pituitary tissue. We characterised the organisation of lactotroph cells in pituitary tissue and found it to be indistinguishable from a random distribution (*Figure 3A*). This does not preclude the presence of direct communication, or of an organised interaction network, between the cells (as previously described [*Hodson et al., 2012*]). We assessed the level of correlation between the fluorescence profiles of individual cells. The increase in PRL gene transcription observed in adult pituitary tissue enabled us to investigate whether cells would coordinate their activity in response to a release from the dopaminergic inhibition that would have occurred in vivo. Similar activity would be expected in vivo when changes in dopaminergic tone facilitate PRL expression and secretion, such as during circadian regulation and lactation (*Ben-Jonathan and Hnasko, 2001*; *Romano et al., 2013*; *Le Tissier et al., 2015*). For any two cells, we calculated the correlation between their fluorescence activity over the time-series as described in *Figure 3B*. Correlation was calculated as a function of the distance between individual cells, and showed that cells within approximately 35 μm (estimates from all datasets were 25–35 μm) of each other were more correlated than cells located further apart. The level of correlation over short distances was also significantly greater than the correlation profile obtained when fluorescence profiles were randomised between the cells, but the positioning information was maintained (*Figure 3B*). This pattern of spatial correlation was not an artefact caused by limitations in image resolution or signal saturation (*Figure 3—figure supplement 1A,B*). Spatial correlation of transcription was maintained when the number of cell pairs in each bin was normalised, indicating that this was not an artefact caused by small numbers of cells in the analysis (*Figure 3—figure supplement 1C*). Spatial correlation also persisted throughout the time-course, indicating that coordination of PRL transcription was an ongoing process and not just a transient phenomenon facilitated by cellular stimulation (*Figure 3—figure supplement 1D*).

The range over which cells would be able to coordinate their transcription activity profiles through a cellular network structure was assessed. We found that cells connected directly, as defined by a threshold distance, had the greatest correlation in activity (*Figure 3—figure supplement 2*), but also that cells connected together via a potential cellular network structure were more correlated than cells that were unconnected (*Figure 3C*). Overall, these data suggest that the mechanism by which lactotroph cells coordinate their functional activity may propagate across the pituitary through a cellular network.

We analysed the timing and direction (up or down) of switches in transcription rate between cells, determined using the stochastic switch model. We tested the hypothesis that cells positioned close together, and that switch activity in the same direction, would do so more synchronously than cells located further apart (scenarios and analysis illustrated in *Figure 4*). Cells located closer together had a tendency to switch activity within a smaller time interval, but only if they switched activity in the same direction (*Figure 4B,C*).

## Patterns of prolactin gene transcription activity and spatial coordination in developing pituitaries

We investigated the involvement of signalling through cell junctions in the coordination of PRL transcription. We tested the hypothesis that the lower lactotroph cell density in developing pituitary tissue would provide less opportunity for cell junction signalling and thereby result in less transcriptional coordination between cells. We characterised the potential for cell junction signalling in E18.5 and P1.5 pituitaries by immunofluorescence (*Figure 5A,B*) and electron microscopy (*Figure 5C–K*). Clustering of lactotroph cells increased in P1.5 pituitaries compared to E18.5 pituitaries (*Figure 5B,D and G*). In the P1.5 pituitary gland lactotroph cells preferentially made contacts with other lactotroph cells, whereas in the E18.5 pituitary gland lactotroph cells were mainly isolated from each other (*Figure 5D,G*). We specifically determined the presence of gap junctions and adherens junctions between lactotroph cells, as these have been implicated in the organisation and function of cell networks in the adult pituitary gland (*Morand et al., 1996*; *Chauvet et al., 2009*; *Kikuchi et al., 2006*). In E18.5 pituitaries, only a small number of adherens junctions could be detected (*Figure 5E*) and the expression levels of the adherens junction proteins (E-, N-cadherin and β-catenin) were very low. E-, N-cadherin, and β-catenin were more abundantly expressed in P1.5

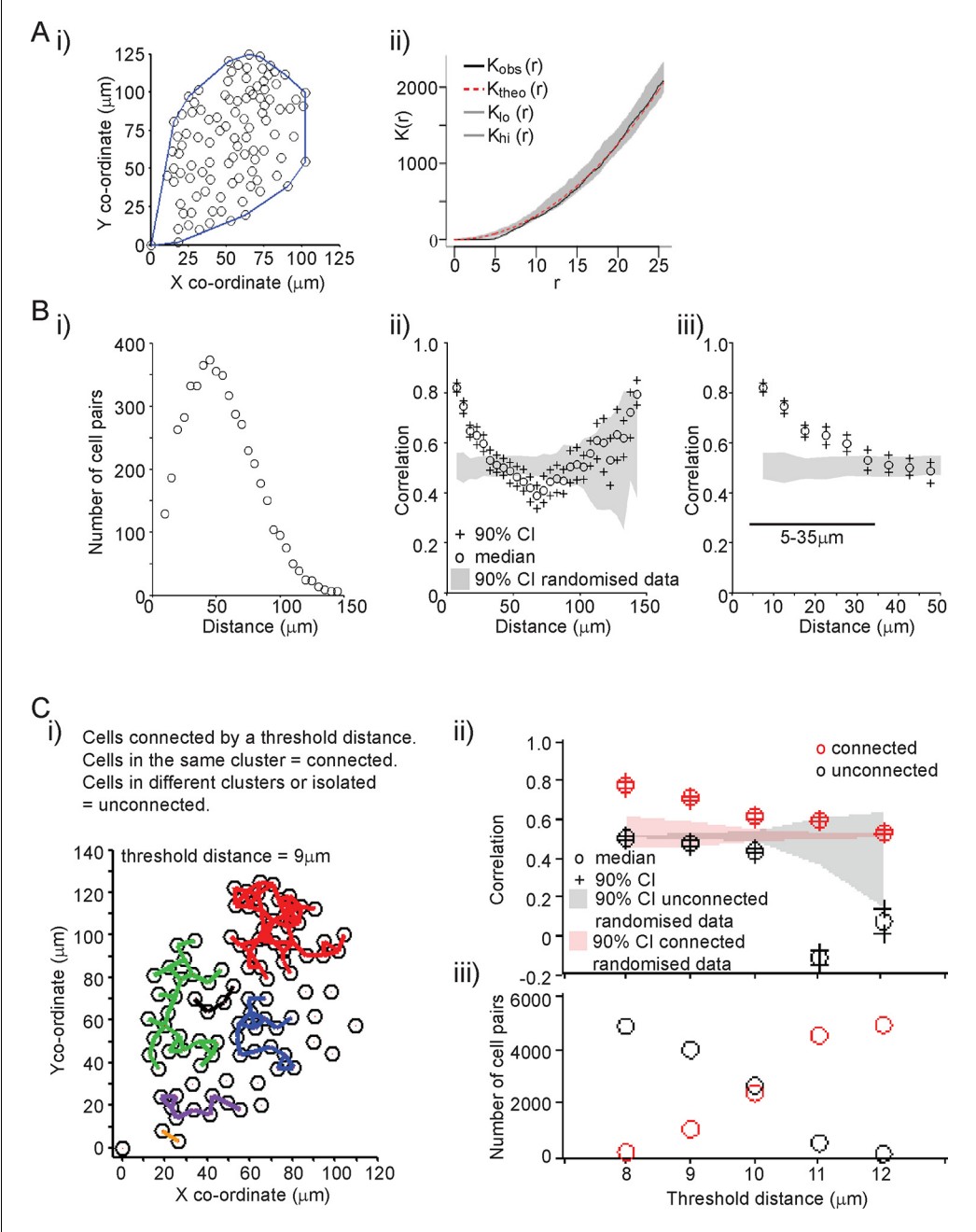

**Figure 3.** Spatial organisation of transcription activity in pituitary tissue. (**A**) Assessment of the spatial distribution of lactotroph cells with PRL gene transcriptional activity. i) The median position of the cells over the time-course along with the field or convex hull occupied by the cells. ii) Ripley's k function was used to test whether the cellular distribution was random. The observed distribution ($K_{obs}$ (r)) deviates from the theoretical prediction ($K_{theo}$ (r)) in the short r range due to the finite size of the cells, above which the cell distribution is found to be indistinguishable from random and within the 95% confidence interval ($K_{lo}$ (r) – $K_{hi}$ (r)). (**B**) Correlation vs distance analysis of cells from adult male pituitary tissue. The correlation between two time series $x_i$ and $y_i$ measured at $N$ times is defined as $N^{-1} \sum_{i=1}^{N} x_i y_i$. i) The distribution of distances between cell pairs. Correlation vs distance plots ii) all cell data and iii) cell pairs within 50 μm; indicate that proximally located cells have more similar activity than cells located further apart. The median and 90% confidence interval (CI) are shown along with the profile obtained from the 90% confidence interval of randomised of cellular transcription patterns. The range shown indicates the distances over which the randomised data were significantly greater (paired t-test, p<0.001) to the non-randomised data. (**C**) Analysis of correlation mediated by a potential lactotroph cell network in adult male pituitary tissue. i) Cell connectivity map with cells classified as connected or unconnected according to the threshold distance used. Connections between cells at a single time point are portrayed. ii) Correlation was calculated between connected (red) and unconnected cells (black) in parallel with randomised data. iii) The number of connected

*Figure 3 continued on next page*

*Figure 3 continued*

and unconnected cell pairs at various threshold distances. Connected cells were more correlated in their transcription activity than unconnected cells. Cell connectivity in ii)-iii) are based on median cell distances across the time-course to take into account cell movement. Data shown are representative of three independent experiments. PRL, prolactin.

The following figure supplements are available for figure 3:

**Figure supplement 1.** Characterisation of spatial organisation of prolactin transcription activity.

**Figure supplement 2.** Correlation of transcription profiles within a cellular network structure.

pituitaries than in E18.5 pituitaries (*Figure 5B*), which coincided with increased numbers of adherens junctions as well as gap junctions and tight junctions (*Figure 5E,H*). In addition to the presence of visually normal junctions (*Figure 5I,J*), we also detected abnormal junctions where cadherin expression at the membrane could be detected but the characteristic thickening of the membrane was absent (*Figure 5K*). These data indicate that, although the potential for communication between lactotroph cells increases during development, cell junction communication in P1.5 pituitaries may still be immature or atypical of the communication that occurs in the adult gland.

Profiles of hPRL-d2EGFP reporter gene activity in developing pituitaries were different to those seen in the adult pituitary. In E18.5 pituitaries, d2EGFP signal was initially very low, but showed a

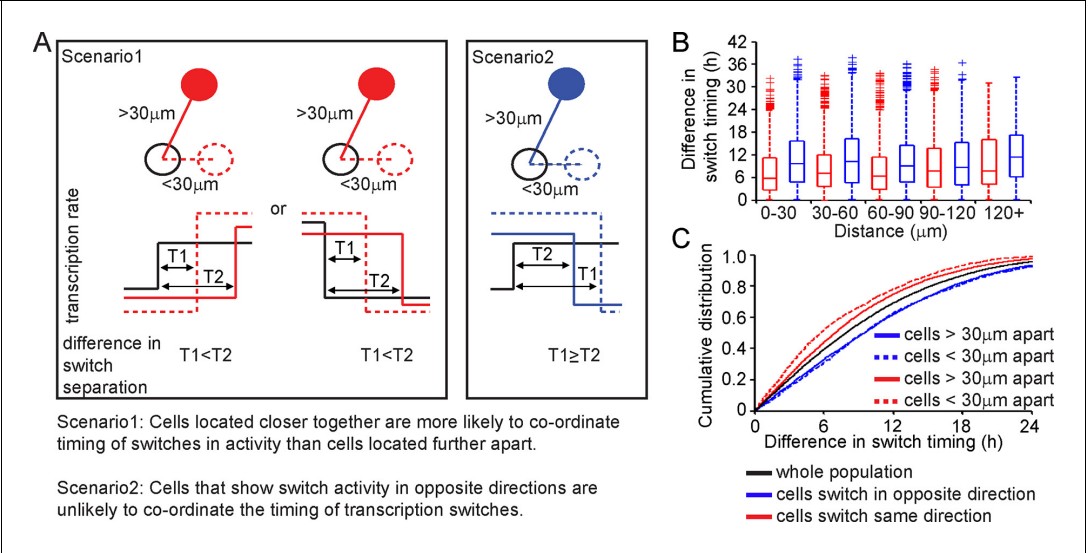

**Figure 4.** Spatial organisation of stochastic switch model derived prolactin transcription dynamics. (**A**) Schematic outlining the hypothesis that was used to assess the spatial organisation of PRL transcription dynamics. The hypothesis was that two cells located closer together will tend to switch transcription in the same direction with more synchronous timing than cells that are located further apart. Moreover, a similar co-ordination in the timing of switches will not be observed if switches occur in the opposite direction. Comparisons are made to the index cell (black). Red denotes cells that switch transcription rate in the same direction, blue denotes cells that switch transcription rate in the opposite direction. T1 is the time interval between cells located within 30 µm (and dashed lines) and T2 is the time interval between cells located more than 30 µm apart (and solid lines). (**B**) Graph showing boxplots of switch timing intervals in cells that switch in the same direction and cells that switch in different directions, binned by the distance between cells. A rising trend is seen in the time interval between transcription rate switch events in cells that switch activity in the same direction (red), but not in cells that switch activity in the opposite direction (blue). Specifically, the median time interval between switch events is smallest in cells that are located within 30 µm and that switch activity in the same direction. Cumulative distributions and significance testing of these differences are shown in (C). All pairwise switches are considered. Boxplots represent the median and interquartile range (IQR), with whiskers drawn 1.5xIQR away from the lower and upper quartile. (**C**) The cumulative distribution of the time interval between switch events shows that cells within 30 µm that switch activity in the same direction (red dashed line) do so within a smaller time frame than cells located greater than 30 µm apart (red solid line), the unsorted population (black) and cells that switch activity in opposite directions (blue dashed and blue solid lines) (confirmed by significant p-value <0.01 of Kolmogorov-Smirnov tests). These were calculated by sampling at random, a pair of possible transcriptional profiles for each pair of cells. Data shown were pooled from three independent experiments.

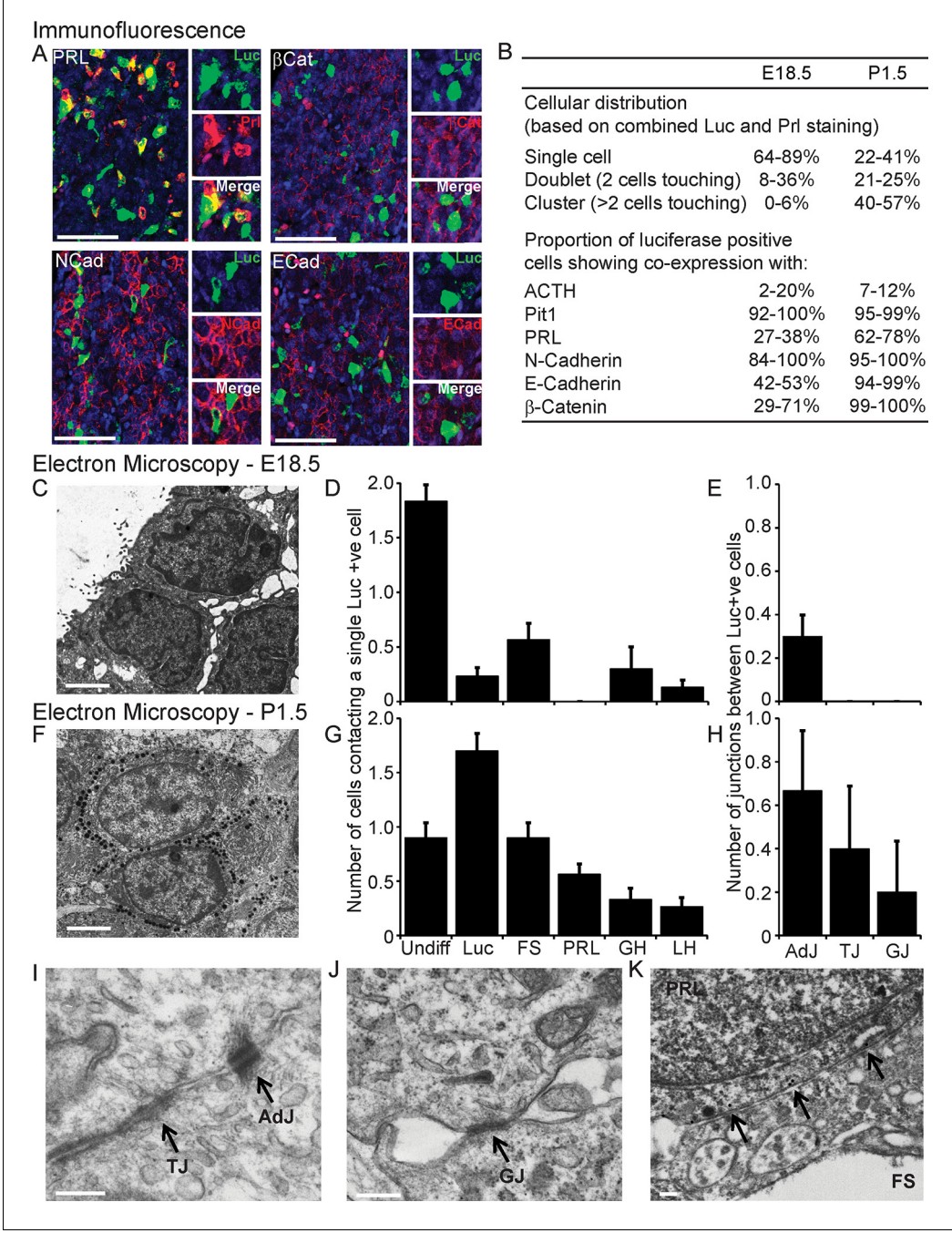

**Figure 5.** Characterisation of cell signalling potential in immature pituitary glands. (**A, B**) Immunofluorescence analysis of lactotroph cell connectivity in developing pituitaries. (**A**) Immunofluorescence images show co-expression of luciferase antibody (green), used to identify lactotroph cells, with PRL, N-Cadherin, E-Cadherin, and β-catenin (red) in paraffin-embedded sections from P1.5 PRL-Luc49 rats. Nuclei were counterstained with DAPI (blue). Right: 8x crop images. Bars in images represent 50 μm. (**B**) Table showing the proportion of lactotroph cell clustering in E18.5 and P1.5 pituitary tissue and the level of co-expression of the luciferase protein and the protein of interest indicated, counted from immunofluorescence images. (**C–K**) Electron microscopy analysis of lactotroph cell connectivity in E18.5 (C-E) and P1.5 (F-K) pituitaries. (**C, F**) Representative image of two luciferase positive cells detected through immunogold labelling of luciferase antibody. Bars represent 1 μm. (**D, G**) Graph of the number of cells contacting an individual luciferase-positive cell. Undiff, undifferentiated cell. Luc, luciferase-expressing cell. FS, folliculostellate cell. PRL, lactotroph. GH, somatotroph. LH, gonadotroph. (**E, H**) Graph quantifying the different types of cell junctions present between luciferase-positive cells. Data in D-E and G-H are represented as

*Figure 5 continued on next page*

*Figure 5 continued*

mean + SEM. (**I, J**) Electron micrograph of a visually normal adherens junction (AdJ), a visually normal tight junction (TJ) and a visually normal gap junction (GJ) in P1.5 pituitaries. Bars represent 200 nm. (**K**) Electron micrograph of an abnormal adherens junction in P1.5 pituitary. Bar represents 200 nm. Information and validation of antibodies used are presented in **Figure 5—figure supplement 1**. SEM, standard error of the mean.

The following figure supplement is available for figure 5:

**Figure supplement 1.** Description and validation of antibodies used.

sustained increase leading to a greater number of cells being detected at the end of the imaging period (**Figure 6A,B**). In P1.5 pituitaries, we again detected pulsatile transcription, with signal increasing after approximately 20 hr of imaging (**Figure 6E,F**). Autocorrelation analysis of hPRL-d2EGFP fluorescence profiles from both E18.5 and P1.5 pituitaries indicated that the majority of cells showed only a single episode of PRL transcription with no dominant period being evident (**Figure 6—figure supplement 1**). Spatial correlation analyses of fluorescence activity, showed that PRL transcribing lactotroph cells in developing pituitaries were more sparsely distributed in comparison to adult pituitaries (**Figure 6C,G**), as expected, and that the correlation between closely localised cells was reduced in comparison to the correlation profiles detected in adult tissues (**Figure 3B** and **6D,H**).

## Comparison of prolactin gene transcription dynamics between developing and adult pituitary tissue

Analysis of fluorescence imaging profiles using the stochastic switch model was used to infer the underlying hPRL transcription dynamics in single cells at different stages of pituitary development. The transcriptional switch data (using data from cells as shown in **Figure 2A** iii) were visualised for all cells in each tissue sample (**Figure 7A**). Inspection of these data suggested that periods of active transcription tend to be of longer duration in adult compared to immature tissue. Direct analyses found no evidence for changes in the distribution of transcription rates at different stages of development (**Figure 7B** and **Figure 7—figure supplement 2A**), even when transcription rates were grouped into low and active states (**Figure 7—figure supplement 1A** and **Figure 7—figure supplement 2B**). However, the number of switches between different rates of activity appeared to be lower in cells in P1.5 pituitary tissue compared to in adult and E18.5 tissues (**Figure 7C**). We also found that pulses of transcription in the highest quartile of transcription rates in E18.5 tissues were clearly of shorter duration than those in the bottom 75%, and in comparison to durations of activity in P1.5 and adult tissue (**Figure 7D** and **Figure 7—figure supplement 2A**). This was also apparent when transcription rates were divided into low and active states and indicates that transcription occurs in a more pulsatile manner in embryonic pituitaries than in more mature tissues (**Figure 7—figure supplement 1B** and **Figure 7—figure supplement 2B**). Where there was more than one switch (and thus the full duration of an interswitch transcriptional state could be determined), the time to the next switch was shorter in immature tissue compared to adult tissue (**Figure 7E**). These data indicate that transcription dynamics are more stable in the adult tissue. No evidence was obtained for spatial coordination of transcription rates in developing pituitaries (**Figure 7—figure supplement 3**).

## Cell communication facilitates spatial coordination of prolactin gene transcription patterns

We next investigated the role of cell junctions in the spatial coordination of transcription in adult pituitary tissue. Trypsin was used as a non-specific protease to digest extracellular proteins and thereby abolish outside-in cell signalling, without tissue disaggregation so that cells were maintained in a tissue environment. Trypsin reduced protein levels of adherens junction proteins E- and N-Cadherin and the gap junction protein Connexin 43, whilst β-catenin, an intracellular component of adherens junctions was unaffected (**Figure 8—figure supplement 1**). Fluorescence profiles of hPRL gene expression from cells in trypsin-treated tissue showed an overall increase in expression levels during the time-course, as did control tissue (**Figure 8A**). Lactotroph cells in trypsin-treated tissue appeared less connected and had a greater intercellular distance than lactotroph cells in untreated

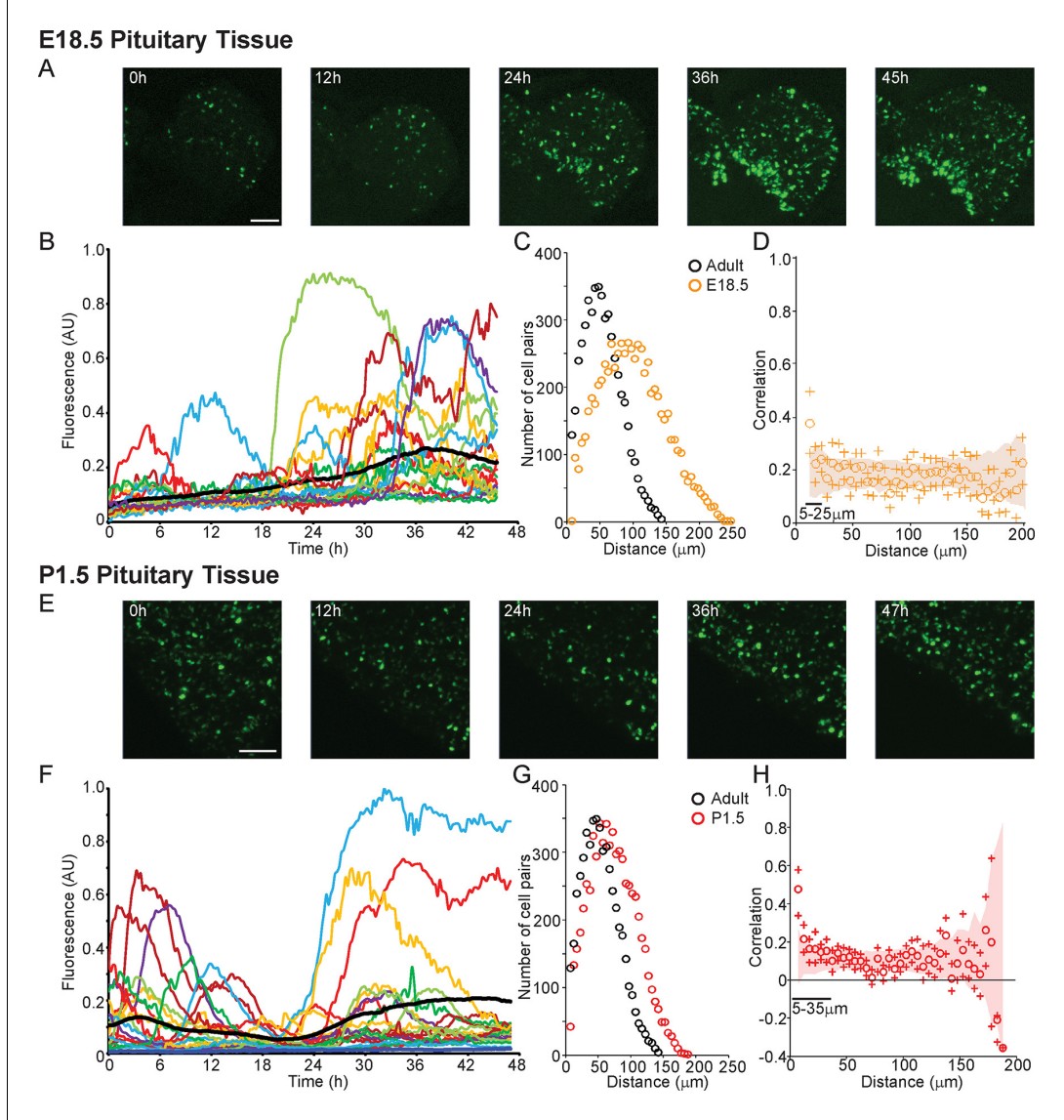

**Figure 6.** Patterns and spatial organisation of prolactin gene transcription activity in immature pituitary tissue. (**A, B**) Activity of the hPRL-d2EGFP reporter construct in single cells in E18.5 pituitary tissue over 46 hr. (**A**) Images of d2EGFP expression in lactotroph cells in E18.5-day-old pituitary tissue (male). (**B**) Fluorescence profiles from 20 individual cells, representative of 136 cells analysed (average intensity, arbitrary units). The black line represents the mean average activity from all the cells analysed (136 cells), the dark blue line represents the background fluorescence intensity (mean from five areas). (**C, D**) Spatial correlation between fluorescence profiles of PRL transcription activity is reduced in E18.5 pituitary tissue in comparison to adult tissue. (**C**) The distribution of cell pairs in E18.5 tissue is compared to the adult cellular distribution. (**D**) Correlation vs distance plots show no large differences between randomised and non-randomised data over short cell-to-cell distances. (**E, F**) Activity of the hPRL-d2EGFP reporter construct in single cells in P1.5 pituitary tissue over 46 hr. (**E**) Images of d2EGFP expression in lactotroph cells in P1.5-day-old pituitary tissue (female). (**F**) Fluorescence profiles from 20 individual cells, representative of 115 cells analysed. Lines are coloured as in (**A**) with the mean representing the activity from all cells analysed. (**G, H**) Spatial correlation between fluorescence profiles of PRL transcription activity was reduced in P1.5 pituitary tissue in comparison to adult tissue. (**G**) The distribution of cell pairs in P1.5 tissue is compared to the adult cellular distribution. (**H**) Correlation vs distance plots show no large differences between randomised and non-randomised data over short cell-to-cell distances. Correlation vs distance plots are shown as described in *Figure 3B*. Images shown are maximum intensity projections of zstacks (E18.5: 165 µm; P1.5: 196 µm). Bar in images represents 100 µm. E18.5 (181 cells) and P1.5 (217 cells) data shown were taken from two independent representative experiments. hPRL, human prolactin.

The following figure supplements are available for figure 6:

**Figure supplement 1.** Autocorrelation analysis of fluorescence profiles from developing pituitary tissue.

**Figure supplement 2.** Comparison of prolactin transcription activity in adult pituitary tissue in different medium.

tissue (*Figure 8B,C*). Spatio-temporal analyses showed a reduction in correlation between the fluorescence profiles of closely localised cells (*Figure 8D*), indicating that cell communication is important for the coordination of PRL transcription dynamics between lactotroph cells in pituitary tissue.

We assessed the transcription response of cells exposed to gap junction inhibitors (18alpha-glycyrrhetinic acid, AGA, and palmitoleic acid, PA [*Juszczak and Swiergiel, 2009*]). Real-time luminometry on pituitary primary cell cultures showed little change in transcription activity at the population level (*Figure 8E*). Fluorescence profiles of hPRL-d2EGFP reporter gene activity in AGA-treated tissue were similar to fluorescence profiles measured in untreated tissue (*Figure 8A,F*) indicating that the gap junction inhibitor AGA had no effect on PRL gene transcription dynamics. AGA also had no effect on the spatial distribution of the reporter-gene-expressing cells (*Figure 8G*). However, correlation between closely localised cells was reduced (*Figure 8H*), indicating a decrease in the coordination of PRL transcription between cells. Overall, these data indicate that cellular communication, at least in part through gap junctions facilitates the coordination of transcriptional activity between lactotroph cells.

## Discussion

Quantitative studies of transcription dynamics have added new levels of complexity to our understanding of gene expression and its regulation. Direct RNA counting techniques (MS2, or RNA-FISH) and the imaging of indirect fluorescence and luminescence reporter gene expression has shown that mammalian gene expression is often pulsatile (or 'bursty') with different genes showing varying characteristics of activity (*Sanchez and Golding, 2013*; *Coulon et al., 2013*; *Suter et al., 2011*; *Spiller et al., 2010*). These studies, performed in single cell systems, raise the question as to how apparently uncoordinated and heterogeneous dynamics enable integrated tissue-level responses to physiological stimuli. We have assessed PRL transcription dynamics within pituitary tissue, utilising newly derived mathematical models to define transcription activity. Cells displayed a continuous distribution of transcription rates with heterogeneous patterns of activity across the cell population. Embryonic pituitary glands displayed shorter durations of high transcription rates compared to adult pituitary tissue, which could reflect epigenetic changes during tissue development. We also characterised the spatial organisation of PRL gene expression within lactotroph cells of the pituitary gland and found evidence for the local coordination of transcription dynamics that is potentially mediated by intercellular signalling providing insights into how transcriptional timing is organised in tissue systems.

Over the past decade, efforts have been made to mathematically model transcription activity to provide a better mechanistic understanding of gene regulation (*Sanchez et al., 2013*; *Larson et al., 2009*). Poissonian distributions of mRNA production, where mRNAs are produced in random, uncorrelated events, have been described (*Zenklusen et al., 2008*; *So et al., 2011*). However, the prevailing model for mammalian transcription dynamics is the Random Telegraph Model, which describes 'bursty' gene expression, where the gene exists in two states, either 'on' or 'off', with transcripts produced at a defined rate in the 'on' period (*Larson et al., 2009*; *Peccoud and Ycart, 1995*). Using binary modelling, we and others have shown that there is a refractory period in the 'off' state, but not in the 'on' state, indicating that there are significant differences between the kinetics of gene activation and inactivation (*Suter et al., 2011*; *Harper et al., 2011*). Binary modelling of transcription dynamics is likely to represent an oversimplification of the true transcription process. Therefore, we developed a stochastic switch model, which allowed us to estimate transcription rates at differing levels and defined the timing of switches between different rates (*Hey et al., 2015*). This model enabled us to characterise transcription activity in cells maintained in tissue, where individual transcriptional states were sustained for long periods and complete cycles of activity were not often detected. Using the stochastic switch model, we found a continuous distribution of transcription rates, that is differential 'on' states, across the cell population, with heterogeneous timing in transcriptional switches between cells. This heterogeneity in transcriptional activity is likely to result from intrinsic factors previously shown to influence PRL transcription (*Harper et al., 2011*), together with extrinsic factors, which include the specialisation of lactotroph cells into distinct subtypes (*Christian et al., 2007*).

To understand the role of stochastic transcriptional processes in tissue biology, it is important to determine how the properties of transcription dynamics are modulated to facilitate changes in gene expression under different physiological states. Modelling of transcriptional activity using a random telegraph process has suggested that levels of gene expression can be controlled through digital

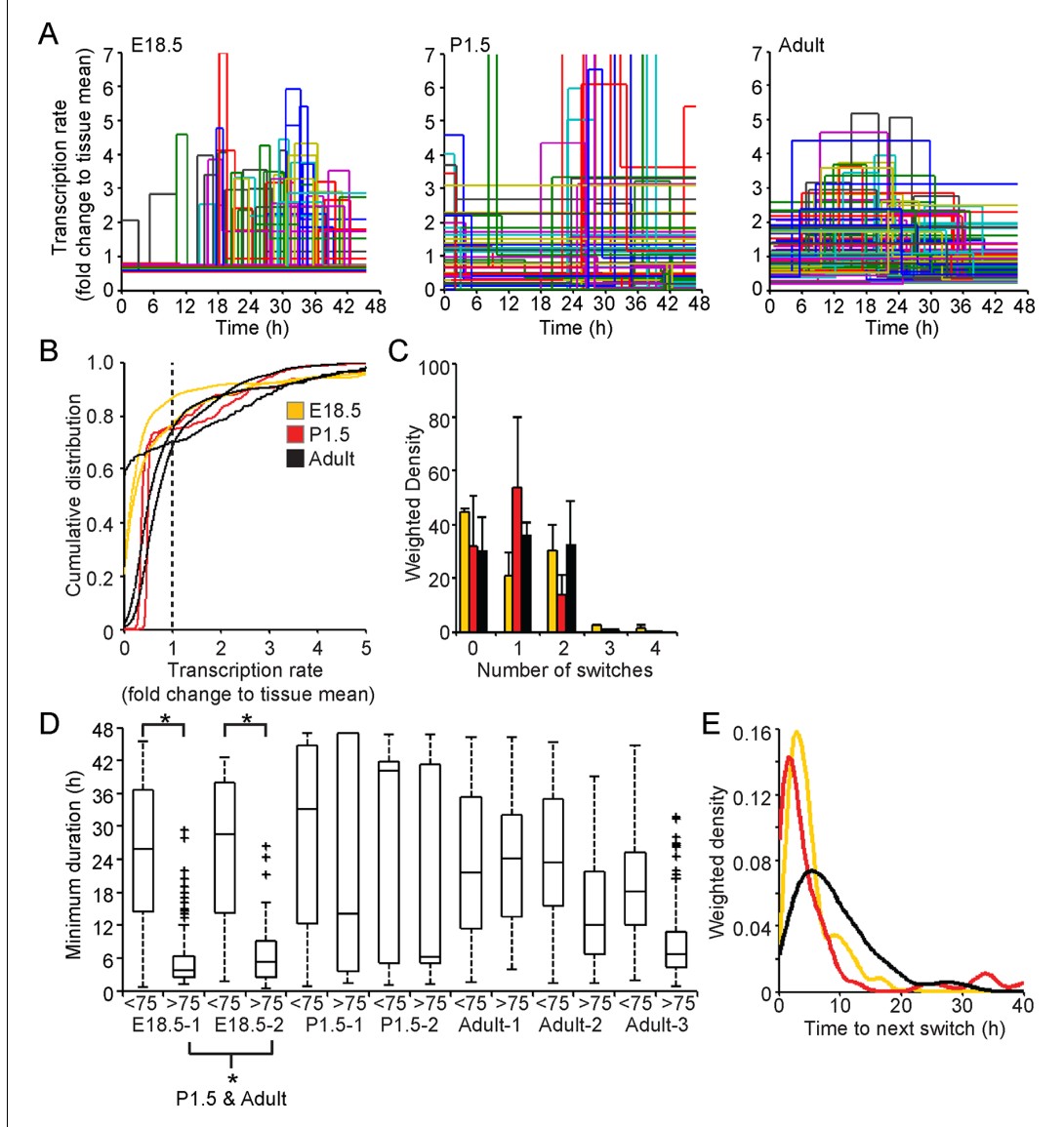

**Figure 7.** Comparison of prolactin transcription dynamics in different pituitary states. (**A**) The stochastic switch model enables comparison of transcription dynamics in different pituitary states. Example plots of transcription rates estimated using the stochastic switch model. Analyses of populations of cells are shown from different states of pituitary tissue development. Each cell is associated with a single transcriptional profile obtained by random sampling of all the possible profiles for that cell (shown as in *Figure 2A* iii, with transcription rate and switch timings depicted). Data shown were from a single experiment representative of independent experiments (i: E18.5 n = 181 cells, ii: P1.5 n = 217 cells, iii: Adult n = 305 cells). (**B–E**) Characterisation of transcription dynamics estimated using the stochastic switch model. (**B**) The cumulative distribution function of the frequency of different transcription rates was estimated using a weighted kernel. No clear difference between pituitary developmental states was evident. (**C**) Weighted histogram of the number of switches for each developmental state. Data, pooled from independent experiments, show the mean + SD, calculated by weighting the number of switches in each profile by the probability of occurrence. P1.5 has significantly fewer switches to Adult and E18.5 (p<0.01 Mann-Whitney U test of the pooled distributions). (**D**) The duration of transcription rates for each tissue state. Transcription rates were binned into quartiles, with the first three lower quartiles grouped together (<75) as one bin and the highest quartile as the other bin (>75). The duration of transcriptional states was calculated for each bin. The duration represents the minimum duration spent in each transcriptional state, as not all transcriptional states were completely observed within the time-frame of the imaging experiment. Both E18.5 datasets showed a significant difference in the duration spent in low transcriptional states (lower 3 quartiles) to high transcriptional states (upper quartile) as did one P1.5 dataset and two Adult datasets. Moreover, the duration spent in the upper quartile of the E18.5 (pooled data) was significantly different to the duration spent in the upper quartile of either the pooled data of P1.5 or Adult. Significance was assessed through the distribution of the p-values calculated from a Mann-Whitney bootstrap (*Figure 7—figure supplement 2*). Boxplots were calculated by a random sampling of the weighted transcription rate distributions and the associated durations. (**E**) The graph shows a kernel density estimate of the time between any two switches using data from panel D, for cells that displayed more than one switch so that the total duration spent in a single transcriptional state was estimated. The inter-switch time in the adult tissues

*Figure 7 continued on next page*

*Figure 7 continued*

is longer than the inter switch times of either the E18.5 or P1.5 tissues. E18.5 = orange, P1.5 = red, Adult = black. All boxplots represent the median and interquartile range (IQR), with whiskers drawn 1.5xIQR away from the lower and upper quartile. SD, standard deviation.

The following figure supplements are available for figure 7:

**Figure supplement 1.** Characterisation of prolactin transcription dynamics.

**Figure supplement 2.** Significance testing of transcriptional state durations.

**Figure supplement 3.** Spatial organisation of transcription switch profiles in developing pituitaries.

responses where the burst frequency or duration is altered, but not the transcription rate (*Larson et al., 2013*). Similarly, digital responses in transcription activity have been detected in single cell systems where the probability that cells are recruited to an expressing population changes under different conditions (*Chubb et al., 2006*; *Walters et al., 1995*; *Kar et al., 2012*). In contrast, different kinetic transcriptional responses including changes to transcription rate and durations of 'on' activity have been found for the connective tissue growth factor (*ctgf*) gene in response to different stimuli (*Molina et al., 2013*), and for housekeeping genes in *Dictyostelium* (*Muramoto et al., 2012*). Overall, the total level of transcription in a given pulse will depend not only on the length of the pulse but also on the rate of transcription during the pulse. Different rates of transcription will depend on levels of RNA polymerase II loading, which may be controlled by different chromatin and promoter states. We observed a continuous distribution of transcription rates within cell populations, indicating that different levels of activity are attainable. However, at the population level similar distributions of activity were detected in all developmental states analysed. Thus, differences in transcription rate contribute to heterogeneous activity at the population level and may be important in maintaining tissue function. In different developmental states, we found changes in the duration of high transcription rates between embryonic and more mature pituitary glands, indicative of a more pulsatile activity in immature tissues. Thus, changes to the duration of activity appear more prominent in facilitating changes in the level of gene expression than changes to transcription rate.

Transcriptional stochasticity within cellular populations may be advantageous in maintaining population fitness to changing environments (*Thattai, 2004*), or facilitate cell fate choices (*Chang et al., 2008*; *Wernet et al., 2006*). However, the role of stochastic gene expression in tissue systems where integrated responses to physiological demand are required is less clear. It has been proposed that heterogeneous responses may facilitate robust tissue-level responses and potentially avoid inappropriate amplification of signals through feedback mechanisms (*Paszek et al., 2010*). In contrast, mechanisms to reduce expression level heterogeneity have been described in processes such as patterning and specification in other species (*Little et al., 2013*; *Raj et al., 2010*). A recent study used single-molecule RNA-FISH at single points in time to define bursting transcriptional behaviour in fixed liver tissue and identified polyploidy as a mechanism to reduce intrinsic variability between cells (*Bahar Halpern et al., 2015*). The pituitary gland is an excellent model system in which to investigate cellular population responses to physiological signals. The gland is composed of multiple cell types that are spatially organised within the pituitary, several of which have been suggested to form interdigitated cellular networks (*Le Tissier et al., 2012*; *Mollard et al., 2012*; *Hodson et al., 2012*; *Fauquier et al., 2001*; *Bonnefont et al., 2005*). Lactotroph cells coordinate their calcium signalling in basal physiological states and more substantially during increased demand such as lactation (*Hodson et al., 2012*). In this study, we have provided a quantitative analysis of lactotroph cell connectivity and shown that PRL transcription is coordinated between lactotroph cells over short distances (25–35 μm) and propagated through a network structure. Transcriptional coordination was actively facilitated by intercellular signalling, and we have shown that this could be via juxtacrine signalling including gap junctions. Intercellular signalling has been shown to be important for coordinating other oscillatory systems such as the circadian clock in the suprachiasmatic nucleus (*Liu et al., 2007*), the somite segmentation clock (*Horikawa et al., 2006*; *Masamizu et al., 2006*), and electrical coupling of and insulin secretion from pancreatic β cells (*Smolen et al., 1993*).

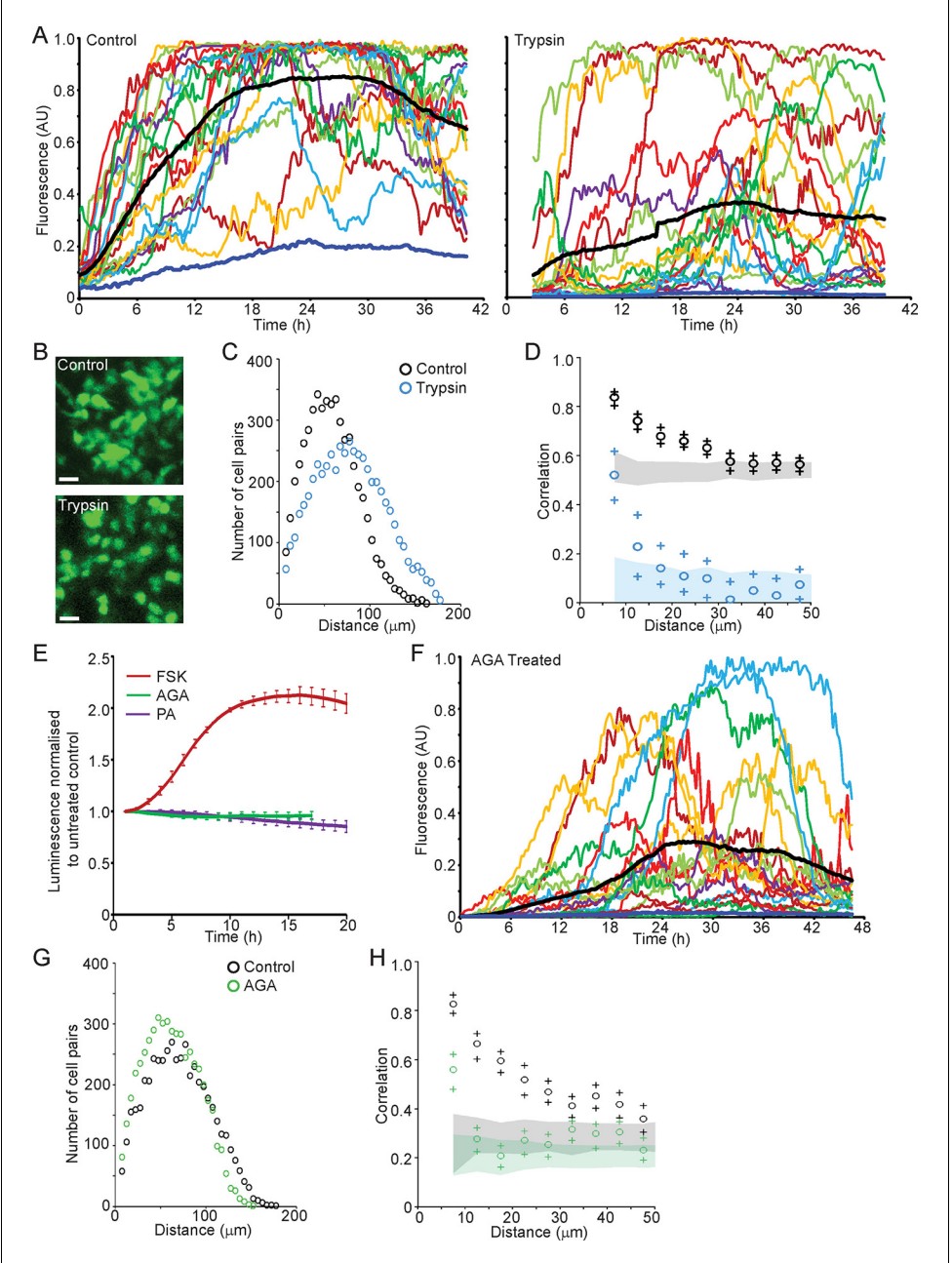

**Figure 8.** Cell communication influences the spatial organisation of prolactin transcription dynamics. (**A**) Comparison of fluorescence profiles of hPRL-d2EGFP reporter gene activity from individual cells in control and trypsin-treated tissue. Cells in trypsin-treated tissue appeared less synchronised over time, but still showed an overall rise in activity as shown by the mean activity (black). The level of background fluorescence is shown in dark blue (mean from five areas). (**B–D**) Spatial correlation between fluorescence profiles of hPRL transcription activity is reduced in trypsin-treated tissue. (**B**) Images of cells within control and trypsin-treated tissue show that the distribution of cells and contacts between d2EGFP-expressing cells appeared altered following trypsin treatment. Bar represents 100 μm. (**C**) The intercellular distance between cells from control and trypsin-treated tissue was calculated as the median distance over the fluorescence imaging time-course (shown in A). (**D**) Correlation vs distance analyses showed a reduction in the difference between non-randomised and randomised data in trypsin-treated tissue compared to control, indicating a reduction in the spatial influence on transcription. (**E–H**) Inhibitors of gap junction signalling were used to assess whether juxtacrine signalling is influential in coordinating PRL transcription activity. (**E**) Real-time luminescence activity from populations of cells in primary cultures show that gap junction inhibitors (18α-glycyrrhetinic acid, AGA, and palmitoleic acid, PA) had little effect on overall PRL gene expression. Forskolin (FSK) was used as a positive control. (**F**) Fluorescence profiles of single cells in AGA treated tissue. Data are represented as described in (A). Transcription activity increased during the time-course (mean activity, black), similarly to control tissue (A). (**G**) The intercellular distance between cells from control and AGA-treated tissue was calculated as the median distance over the fluorescence imaging time-course (shown in A,F). (**H**) Correlation vs distance analyses showed a reduction in the difference between

*Figure 8 continued on next page*

*Figure 8 continued*

non-randomised and randomised fluorescence profiles in AGA treated tissue compared to control tissue indicating a reduction in the spatial coordination of transcription. Correlation vs distance plots are shown as described in *Figure 3B*.

The following figure supplement is available for figure 8:

**Figure supplement 1.** Effect of trypsin on cell junction proteins.

The global picture that arises is that transcription is highly stochastic but has some coordination of bursting at distances up to approximately 35 μm in adult pituitary tissue, but not at greater distances. In contrast there was no coordination at any intercellular distance in earlier developmental states. The limited short distance coordination between lactotroph cells in the adult tissue is not sufficiently strong to lose the key characteristic of cell-to-cell heterogeneity. However, it can be hypothesised that the global system of short range cell-to-cell communication may stabilise longer term changes in the expression level of the tissue, such as those associated with the oestrus cycle or lactation. Thus far the gland as a whole prolactin transcription is essentially random in that for the vast majority of cell pairs, the temporal pattern of their transcription is uncoordinated. Therefore, the law of large numbers guarantees stable long-term results in terms of the global response.

Our work addresses how an endocrine tissue, such as the pituitary gland, generates a controlled output from a diverse set of intermingled cell types. The acute, medium, and long-term outputs of endocrine cells must be regulated across different time scales and to diverse environmental signals, while achieving accurate control of hormone expression. A key question has been whether cells behave similarly to each other or whether they operate in a heterogeneous autonomous fashion. Previous data have suggested the latter in isolated cells and cell lines. Our data now indicate that developing adult tissue structure exerts a coordinating effect on cell behaviour (see *Figure 9A*). Our observation that cells display multi-state transcriptional behaviour ('off', 'primed' and 'on' at various levels; outlined in *Figure 9A*), as opposed to simple binary 'on'/'off' behaviour, is suggestive of mechanisms that tune the overall output from the gland through the generation of graded responses. Changes in the duration of particular transcription states, as we observed with shorter 'on' periods in immature pituitary glands, also help to refine the overall output from the gland. A simulation (*Figure 9B*) shows how decreasing the duration of the 'primed' period allows a population of cells to switch to a new stable state of mRNA production. Such modification of activity may facilitate the differentiation of the tissue response without a risk of overshooting behaviour. This offers a new mechanism to explain how tissues integrate different signals with varying durations into appropriate responses. Examples of dynamic control of the pituitary gland include the acute suppression or activation by hypothalamic regulators, circadian response, and long-term behavioural changes through development, puberty, pregnancy, and ageing.

In summary, the results presented provide a quantitative framework of cellular transcriptional activity within living tissue. We demonstrate that there is local transcriptional coordination between prolactin-expressing cells in adult, but not in developing pituitary tissue. We have described quantitative analyses of transcription kinetics in different developmental states of the pituitary gland. In all states, we showed that transcription occurs at different levels of activity separated by statistically definable switches. We also detected changes in transcription dynamics between different stages of pituitary development. In the adult tissue, the cells show longer lived active states, associated with an altered environment in which there is greater cell-to-cell connectivity. We have identified potential mechanisms by which the combined dynamics of single cells within a tissue may achieve both acute and long-term functional adaptation of the expression of secreted regulatory proteins.

## Materials and methods

### Animals

Animal studies were performed under UK Home Office License and subject to local ethical committee review. The generation and characterisation of Fischer 344 transgenic rats with the luciferase (PRL-Luc49) or destabilised GFP (PRL-d2eGFP455) reporter genes under the control of the hPRL locus has

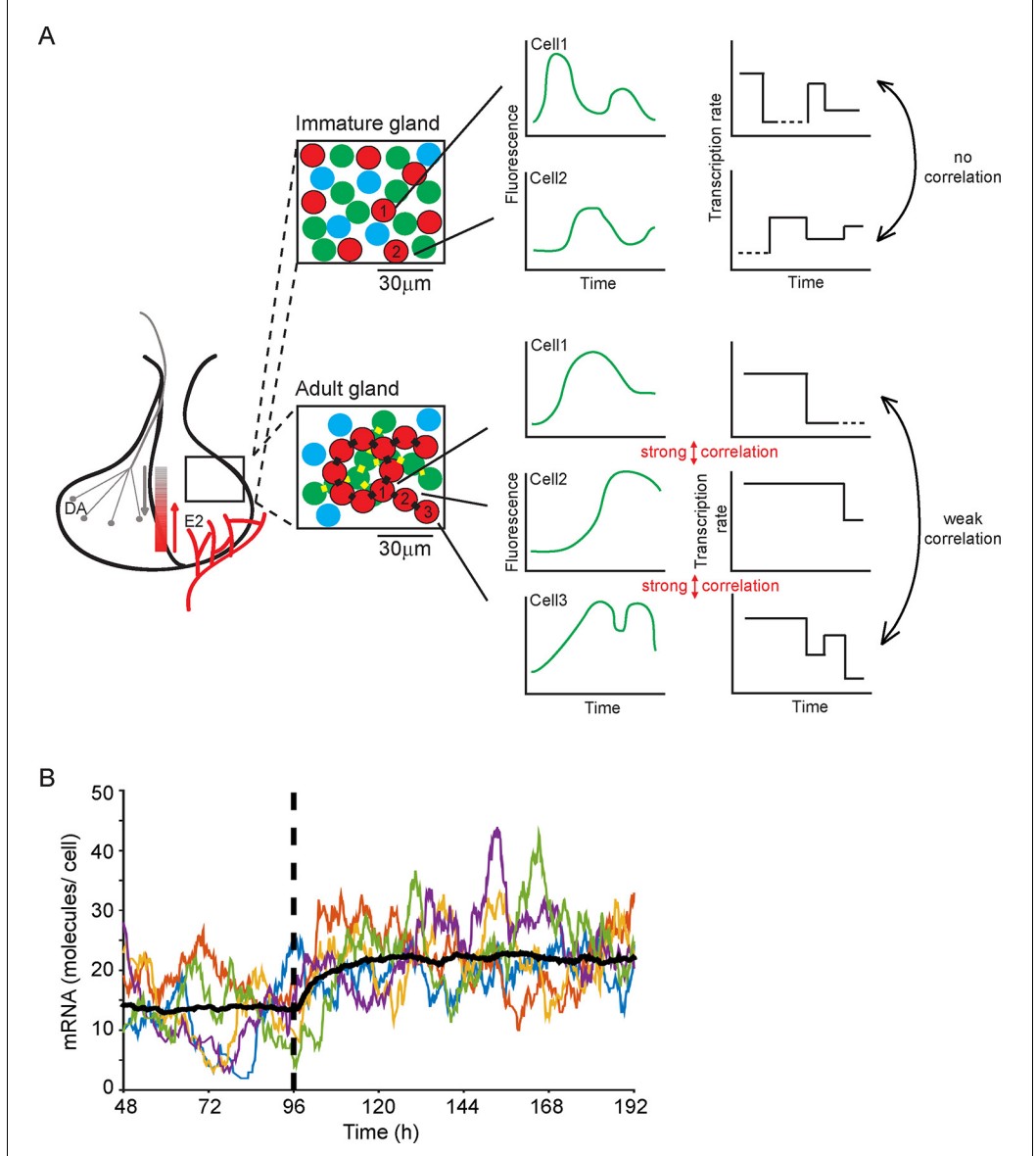

**Figure 9.** Multi-state transcription dynamics achieve robust tissue level responses. (**A**) Organised cell networks and gap junction cell-cell signalling in adult pituitary glands (lower panels), but not in immature pituitary glands (upper panels), leads to local correlation of transcription between cells. Across a large population of cells transcription dynamics are essentially random and uncoordinated. Multi-state transcription dynamics ('off', 'primed' and 'on' at various levels) contribute to transcriptional heterogeneity between cells as the time spent in each state is exponentially distributed. The schematic indicates putative regional signalling and signalling gradients across the pituitary, for example from hypothalamic dopamine (DA) and systemic estradiol (E2). Segments of pituitary tissue are represented with lactotroph cells (red) connected via gap junctions (black rectangles). Examples of single cell transcription profiles (left plots) and their relation to multi-state transcription dynamics (right plots), with 'off', 'primed' (dashed lines) and different levels of transcription in the 'on' state are shown. (**B**) Simulation of the telegraph model with priming demonstrates how decreasing the 'primed" period (at the dashed line) gives rise to a rapid robust response across a population of cells showing heterogeneous activity. The simulation was performed on fluorescence profiles from 100 cells, with the mean of these responses shown (black line) along with 5 representative cell responses (coloured lines).

been described previously (*Semprini et al., 2009*). The BAC transgenes are integrated into the genome at a single site with high copy number in the PRL-Luc49 line and low copy number (≤5) in the PRL-d2eGFP455 line (*Harper et al., 2010*; *Semprini et al., 2009*). Animals were housed in humidity- (50 ± 10% ) and temperature- (20 ± 1°C) controlled conditions in a 12 hr light:dark cycle with food (rat and mouse standard diet; Special Diet Services, Witham, UK) and water ad libitum. Timed matings

were set-up with transgenic males and wild-type females with the detection of a vaginal plug the morning after mating considered as E0.5. Animals were genotyped using DNA extracted from ear biopsies (adults) or tail biopsies (fetal and neonates). Young animals were sexed using PCR conditions described in (*Featherstone et al., 2011*). Genotyping was performed on PRL-Luc49 DNA as described in (*Featherstone et al., 2011*). Genotyping of PRL-d2eGFP455 DNA was performed with the same conditions as for PRL-Luc49 DNA but with d2eGFP primers substituted for luciferase primers; d2eGFP forward, 5'-GACGACGGCAACTACAAGAAC -3' and d2eGFP reverse, 5'-ACTCCAG-CAGCACCATGTGAT -3'. Animals were sacrificed by a schedule 1 method (exposure to a rising concentration of $CO_2$ followed by cervical dislocation) followed by resection of pituitary glands.

## Preparation and culture of pituitary tissue

Coronal slices of adult pituitary glands (300 µm) were cultured on 0.4-µm filter stages (Greiner Bio-One, UK) in 35-mm glass-coverslip-based dishes with access to air and medium (DMEM + 4.5 g/l glucose, 10% (v/v) FBS, 1 mM sodium pyruvate, 100 U/ml penicillin, 100 µg/ml streptomycin, and 2 mM ultraglutamine) (*Figure 1A*). Whole pituitary glands from fetal (E18.5) (n = 2) or neonatal (P1.5) (n = 2) animals were treated as described in (*Featherstone et al., 2011*) except that luciferin was omitted from the medium and pituitaries were cultured on filter stages as described above. Adult pituitary tissue was either untreated (n = 3), treated with Trypsin (0.1% (w/v) Trypsin (Sigma UK), 0.0045% (w/v) DNase I, 0.325% (w/v) BSA in HBSS) (n = 2), or AGA (20 µM in 10% FBS medium) (n = 2) for 2 hr at 37°C prior to imaging.

## Fluorescence confocal imaging of pituitary tissue

Pituitaries were imaged using Carl Zeiss laser scanning confocal microscopes (LSM): Pascal, 710 and 780, maintained at 37°C in PeCon XL incubators (PeCon, Germany) with a humidified atmosphere of 95% air and 5% $CO_2$ and with a Fluar 10X magnification 0.5 NA objective. Excitation of d2EGFP was performed using an argon ion laser at 488 nm with emitted light captured through appropriate filters or a selected portion of the spectrum. All imaging was acquired as z-stacks with images captured in 15 min intervals for 48 hr in basal culture medium and then for 24 hr following forskolin stimulation (Adult: 5 µM or Immature Pituitaries: 1 µM). Fluorescence from tissues was analysed as maximum intensity projections using ZEN 2010b (Zeiss, UK) or CellTracker software (http://www2.warwick.ac.uk/fac/sci/systemsbiology/staff/bretschneider/celltracker/).

## Analysis of imaging data

Spatio-temporal analyses were performed by measuring the fluorescence intensity from all cells within an area, encompassing approximately 100 cells. The area analysed was always taken from the lateral edge of the pituitary to minimise differences between cellular activities that may exist across the gland. Regions of interest were drawn around cells and mean intensity data collected using CellTracker software. Cell areas analysed were consistent with the size of a typical eukaryotic cell (*Figure 1—figure supplement 1*). Matlab R2014a software (MathWorks, UK) including Bioinformatics and Statistical toolboxes, or the R programming language (www.r-project.org) were used for mathematical analyses. In all analyses, the positioning of cells was taken as the median x,y coordinates of the centroid. The spatial distribution of cells was tested using a 2D Poission process and Ripley's K function (*Ripley, 1976*). Correlation analyses of transcription patterns from fluorescence data were performed in two ways: either using the Euclidean distance between cells or by a network analysis. Correlation based on the Euclidean distance was performed in two parts. In the first part, cell pairs were partitioned into bins according to the distance between the cells starting at 5 µm with 5 µm intervals. Correlation analysis was performed 99 times for each bin, with n cell pairs sampled with replacement, and with the 5 and 95 percentiles reported. In the second part, n time-series are randomly permuted while the spatial information is unaltered following which correlation coefficients are calculated as described above. A paired t-test was used to identify significant differences between space-time correlations from randomised and non-randomised data. Network correlation analysis was performed similarly to Euclidean distance correlation analyses except that cell pairs were divided into two bins, connected and unconnected, which was defined in a network approach as follows. Cells were assumed to be circular with the same diameter (D). If the distance between two cells was smaller than D, they are considered to be connected. Furthermore cell connectivity is transitive so that if cells a and b are connected and so are cells b and c, then cells a

and c are considered to be connected being part of a cell cluster, even if separated by a distance greater than D. Analysis is performed with a varying D.

The inference of transcription rates from fluorescence data using the stochastic switch model was performed as described (*Hey et al., 2015*), using the following stochastic reaction network:

$$\emptyset \xrightarrow{\beta(t)} \text{mRNA}$$
$$\text{mRNA} \xrightarrow{\delta_m} \emptyset$$
$$\text{mRNA} \xrightarrow{\alpha} \text{mRNA} + \text{Protein}$$
$$\text{Protein} \xrightarrow{\delta_p} \emptyset \qquad\qquad (*)$$

where $\delta_m$ and $\delta_p$ denote the degradation rates of reporter mRNA and reporter protein respectively, $\alpha$ denotes the rate of translation and $\beta(t)$ denotes the time varying rate of transcription. Specifically, the transcription function is given by,

$$\beta(\text{t}) = \beta_i \text{ for } t \in [s_{i-1}, s_i) \text{ for } i = 1, ..., K,$$

where K is the number of transcriptional switches, occurring at times $s_1$, $s_2$, ..., $s_K$ and $\beta_1$, $\beta_2$, ..., $\beta_K$ are the corresponding transcriptional rates. We impose no restriction to the form of the transcriptional levels but note that the conventional binary switch behaviour can be seen as a specific example where $\beta_i = \beta_{LOW}$ if the gene is inactive in the time period $[s_{i-1}, s_i)$ or $\beta_i = \beta_{\text{HIGH}}$ if the gene is active.

Assuming light intensity measurements are related to reporter protein levels through the equation,

$$Y(t) = \kappa \ P(t) + \varepsilon(t),$$
$$\varepsilon(t) \sim N(0, \sigma^2),$$

inference is performed through the linear noise approximation to the system in (*) to obtain the posterior transcriptional function for each single cell. To ensure model identifiability, we impose informative prior distributions about the degradation parameters, obtained from independent experiments as described in (*Finkenstädt et al., 2013*). In addition, we specify a hierarchical framework over each dataset, as individual parameters are unlikely to change substantially. We confirmed that the degradation parameters in tissue samples were similar to estimates obtained using cell lines.

In order to estimate both the number and positioning of transcriptional switches, we employ a reversible jump Markov chain Monte Carlo (MCMC) algorithm (*Green, 1995*). Consequently, the posterior distribution consists of all possible transcriptional profiles. In order to extract the information regarding the estimated transcriptional dynamics, the posterior samples go through a post-processing procedure outlined below:

1. First, we fit a parametric model to the marginal posterior switch distribution (as described in [*Jenkins et al., 2013*]). Specifically, a Gaussian mixture model is fitted to the marginal posterior distribution of the possible switch times.
2. Since this marginal model does not take into account the co-occurrence of switches, we then extract all possible sub-models. For example, if the marginal posterior has two possible switch positions (as shown in *Figure 2A*) the sub-models will consist of a zero switch model, two mutually exclusive one switch models and the two switch model. Counting the frequency with which each of the sub-models was sampled in the MCMC, we can associate a weight or probability to each sub-model.

Thus, this post-processing procedure associates each single cell to a set of mutually exclusive transcriptional profiles. The analysis presented in the main paper has been calculated from the set of all possible transcriptional profiles, weighted by their probability of occurrence. Data of d2EGFP fluorescence measured in single cells and subsequent stochastic switch modelling of transcription activity have been deposited in the Dryad data repository (*Featherstone et al., 2015*)

## Immunofluorescence of immature pituitary glands

Immunofluorescence was performed on Prl-Luc49 immature pituitary glands maintained in situ to enable sectioning in the coronal orientation. Prl-Luc49 rats were used in immunofluorescence and electron microscopy analyses as multiple copies of the Prl-Luciferase transgene enabled the

detection of low expressing PRL cells. Exposed pituitaries were fixed in Bouin's solution (Sigma, UK) for 24 hr, washed with water and 70% ethanol and then embedded in wax and 5-μm sections cut and mounted onto poly-L-lysine slides (Thermo Fisher, UK). Immunofluorescence staining was performed as previously described (*Semprini et al., 2009*) except that slides were pretreated with 3% (v/v) $H_2O_2$, in methanol for 30 min following antigen retrieval. Pituitaries were stained with mouse anti-luciferase (*P. pyralis*) antibody (Life Technologies, UK); co-stained either with: mouse anti- ACTH (Abcam, UK), goat anti-Pit1 (Santa Cruz, UK), rabbit anti- PRL (R51, A McNeilly, Edinburgh, UK), mouse anti-E-Cadherin (BD Biosciences, UK), rabbit anti- N-Cadherin (Calbiochem, UK), mouse anti-β-Catenin (BD Biosciences, UK) and counterstained with DAPI. Briefly, all slides were blocked with blocking buffer (10% (v/v) donkey serum, 5% BSA (w/v) in PBS) and stained with primary antibody in blocking buffer at 4°C overnight. The primary antibody was detected with anti-mouse-HRP (Vector Labs, UK) and Tyramide Signal Amplification- FITC (PerkinElmer, UK). Slides were boiled in sodium citrate (10 mM pH6) for 2 min and left to stand in hot buffer for 30 min before sequentially blocking with biotin, streptavidin (Vector Labs, UK) and blocking buffer and application of second primary antibodies at 4°C, overnight. Detection of the second primary antibodies was with either anti- rabbit biotin, anti- goat biotin, or anti-mouse biotin (Vector Labs, UK) and subsequently with streptavidin- alexa546 (Life Technologies, UK). Specificity of antibody labelling was confirmed in negative control sections in which the primary antibody was replaced with the appropriate non-immune serum. Slides were examined and images taken using a Zeiss Excitor confocal microscope with Fluar 20X magnification 0.75 NA objective. For each developmental stage three pituitaries were analysed (two males and one female were analysed from two independent litters) with co-localisation between antibodies counted across at least one whole pituitary slice. No sexual dimorphism was detected.

## Electron microscopy of developing pituitary glands

Electron microscopy was performed on dissected immature pituitaries, which were processed and immunogold-labelled as previously described (*Abel et al., 2013*). Briefly, the tissue was contrasted with uranyl acetate (2% (w/v) in distilled water), dehydrated in methanol and embedded in LR Gold resin. Ultrathin sections (50–80 nm) were prepared using a Reichart-Jung ultracut microtome and mounted on nickel grids (Agar Scientific, UK). For identification of adherens junctions, sections were immunogold-labelled for E-cadherin (mouse anti- E-Cadherin: BD Biosciences, UK) or β-catenin (mouse anti-β-catenin: BD Biosciences, UK). In order to identify lactotroph cells, sections were immunogold labelled for PRL (rabbit anti- PRL R51: A McNeilly, Edinburgh, UK) and to identify luciferase-positive cells, sections were immunogold labelled for luciferase (mouse anti-luciferase: LifeTechnologies, UK). Sections were counterstained with lead citrate and uranyl acetate and examined on a JOEL 1010 transmission electron microscope (JOEL USA, USA). Specificity of antibody labelling was confirmed in negative control sections in which the primary antibody was replaced with the appropriate non-immune serum. For each pituitary (n = 3), 10 luciferase-immunogold labelled cells were identified and the adjacent cells and intercellular junctions identified on the basis of morphological criteria and counted.

## Primary cultures and live-cell luminometry

Primary cultures of adult female pituitary glands were prepared as described (*Featherstone et al., 2011*). Cells were resuspended in medium (DMEM + 4.5 g/l glucose, 10% FBS, 1 mM sodium pyruvate, 100 U/ml penicillin, 100 μg/ml streptomycin, and 2 mM ultraglutamine) and plated ($1 \times 10^5$/ well) in white plastic 96-well plates pre-treated with poly-L-Lysine at. Cells were allowed to recover for 24 hr (37°C, 5% $CO_2$) after which the medium was replaced with serum-deprived medium (DMEM + 4.5 g/l glucose, 0.25% (w/v) BSA, 1 mM sodium pyruvate, 100 U/ml penicillin, 100 μg/ml streptomycin and 2 mM ultraglutamine) supplemented with 1 mM luciferin (Biosynth, Switzerland) and cells were incubated for a further 24 hr (37°C, 5% $CO_2$). Cells were then treated with either: DMSO (control), 20 μM α-glycyrrhetinic acid (Sigma, UK), 50 μM Palmitoleic acid (Sigma, UK), or 5 μM forskolin (Sigma, UK). Cell responses were measured using the FLUOstar Omega $CO_2$- and temperature-controlled luminometer plate reader (BMG Labtech) with photon counts collected for 10s per well every 15 min for 24 hr. Three independent cultures were analysed with 3-4 replicates per treatment group.

## Analysis of pituitary slice luminescence activity using photon multiplier tubes

Pituitary tissue slices from Prl-Luc49 adult male rats were prepared as described in 'Preparation and culture of pituitary tissue'. Slices were cultured on filter stages in 35-mm dishes in either FBS supplemented recording medium (DMEM + 4.5 g/l glucose, 10% (v/v) FBS, 1 mM sodium pyruvate, 100 U/ml penicillin, 100 µg/ml streptomycin, and 2 mM ultraglutamine, 10 mM HEPES, 1 mM luciferin) or rat serum supplemented recording medium (DMEM and F12, 50% (v/v) serum from sacrificed animal, 100 U/ml penicillin, 100 µg/ml streptomycin, 2 mM ultraglutamine, 10 mM HEPES, 1 mM luciferin) in a closed system at 37°C, to show that differences in transcriptional activity seen between adult and immature tissues was not due to the medium used (*Figure 6—figure supplement 2*). Bioluminescence emissions were recorded by photon multiplier assemblies (Hamamatsu Photonics, UK) with counts collected over a 1-min period. The mean and standard deviation were calculated from data collected over hourly periods. Two independent adult male rat pituitaries were analysed with each condition performed in triplicate.

## Western blotting

Pituitary tissue slices were prepared and either untreated or treated with trypsin for 2 hr at 37°C, as described in 'Preparation and culture of pituitary tissue'. Slices were washed three times in medium (DMEM + 4.5 g/l glucose, 10% (v/v) FBS, 1 mM sodium pyruvate, 100 U/ml penicillin, 100 µg/ml streptomycin and 2 mM ultraglutamine) and then cultured on filter stages as described previously at 37°C, 5% $CO_2$ for either 0, 24, or 48 hr. Slices were lysed in RIPA buffer (50 mM Tris-HCl pH8, 150 mM NaCl, 1% (v/v) Nonidet P40, 0.5% (w/v) sodium deoxycholate, 0.1% (w/v) SDS) with Complete Mini EDTA-free Protease Inhibitors (Roche, UK). Three independent cultures of trypsin-treated and untreated tissue were analysed by western blot. Lysates prepared from cell lines or whole tissue samples were generated by lysis in RIPA buffer with Complete Mini EDTA-free Protease Inhibitors (Roche, UK) following washing with cold HBSS. All samples were subjected to SDS-PAGE (10%) before transfer to nitrocellulose membrane. Primary antibodies (rabbit anti- α-tubulin (Proteintech, UK), mouse anti-E-Cadherin (BD Biosciences, UK), mouse anti-β-catenin (BD Biosciences, UK), rabbit anti-N-Cadherin (Calbiochem, UK), mouse anti-connexin43 (Santa Cruz Biotechnology, USA) were applied overnight at 4°C in blocking buffer (Tris buffered saline, 5% (w/v) dried milk, 0.1% (v/v) Tween20), and species-specific horseradish peroxidase conjugated secondary antibodies (GE Healthcare, UK) were applied for 1 hr at room temperature. Staining was detected with Clarity Western ECL Substrate (Biorad) using Biomax XAR film (Kodak, UK). Protein levels were standardised to α-tubulin expression following quantification of protein expression by determining the relative band size using ImageJ software. For further information on antibodies used see *Figure 5—figure supplement 1*.

## Acknowledgements

We thank M Belle, C Harper, H Piggins, L Scott and S Semprini for assistance with the work; P LeTissier for discussions and advice on the manuscript; staff of the University of Manchester Animal Facility and P Walker of the Core Histology Facility for technical assistance. The Centre for Endocrinology and Diabetes is supported by the Manchester Academic Health Sciences Centre (MAHSC) and the NIHR Manchester Biomedical Research Centre. Hamamatsu Photonics and Carl Zeiss Limited provided technical support to the Systems Biology Centre. We also acknowledge support from British Heart Foundation Centre of Research Excellence Award.

## Additional information

### Funding

| Funder | Grant reference number | Author |
| --- | --- | --- |
| Wellcome Trust | Programme grant: WT091688 | Julian RE Davis Michael RH White David A Rand |

| Engineering and Physical Sciences Research Council | PhD Studentship grant: ASTAA1112.KXH | Kirsty Hey |
| --- | --- | --- |
| Medical Research Council | MR/K015885/1 | David G Spiller<br>Michael RH White<br>Julian RE Davis |
| Biotechnology and Biological Sciences Research Council | BB/K003097/1 | David A Rand<br>Michael RH White |

The funders had no role in study design, data collection and interpretation, or the decision to submit the work for publication.

### Author contributions

KF, Conceived and designed the experiments. Performed the experiments. Analysed the data. Wrote the paper. Initiated and directed the project. Contributed to discussion and critical analysis of the data; KH, Analysed the data. Designed and developed the statistical algorithms. Wrote the paper. Contributed to discussion and critical analysis of the data; HM, Analysed the data. Designed and developed the statistical algorithms. Contributed to discussion and critical analysis of the data; AVM, ALP, JJM, Contributed to discussion and critical analysis of the data; JW, HCC, Performed the experiments. Contributed to discussion and critical analysis of the data; DGS, Contributed reagents/ materials/ analysis tools. Managed the Systems Biology Centre. Contributed to discussion and critical analysis of the data; ASM, Contributed reagents/materials/analysis tools. Contributed to discussion and critical analysis of the data; BFF, Designed and developed the statistical algorithms. Planned and led the mathematical analysis. Contributed to discussion and critical analysis of the data; DAR, Wrote the paper. Planned and led the mathematical analysis. Initiated and directed the project. Contributed to discussion and critical analysis of the data; MRHW, Wrote the paper. Initiated and directed the project. Contributed to discussion and critical analysis of the data; JRED, Conceived and designed the experiments. Wrote the paper. Initiated and directed the project. Contributed to discussion and critical analysis of the data

### Ethics

Animal experimentation: All procedures were subject to local ethical committee review at the University of Manchester and were approved by the UK home office (PPL 40/3296), subject to the restrictions and provisions contained in the Animals (Scientific Procedures) Act 1986.

## Additional files

### Major datasets

The following datasets were generated:

| Author(s) | Year | Dataset title | Dataset URL | Database, license, and accessibility information |
| --- | --- | --- | --- | --- |
| Featherstone K, Hey K, Momiji H, McNamara AV, Patist AL, Woodburn J, Spiller DG, Christian HC, McNeilly AS, Mullins JJ, Finkenstadt B, Rand DA, White MRH, Davis JRE | 2015 | Data from: Spatially Coordinated Dynamic Gene Transcription in Living Pituitary Tissue | http://dx.doi.org/10.5061/dryad.s04bb | Available at Dryad Digital Repository under a CC0 Public Domain Dedication |

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
