## [Decision Letter]

Thank you for submitting your work entitled "Spatially Coordinated Dynamic Gene Transcription in Living Pituitary Tissue" for peer review at *eLife*. Your submission has been favorably evaluated by Jim Kadonaga (Senior editor), a Reviewing editor, and two reviewers.

The reviewers have discussed the reviews with one another and the Reviewing editor has drafted this decision to help you prepare a revised submission.

The Reviewers find the work of interest, in particular that the spatial relationship of cells within a tissue may lead to a common basis for expression. However they feel that there are some obscure analyses and the text in ambiguous in parts. One reviewer "found the work deficient in terms of the quantitative and theoretical analysis". The other thought that" the primary contribution of this paper may be the relationship between this stochastic process and the integrated production from the tissue" but that this was not clearly developed. In both cases, some extensive revisions - in particular, more rigorous text and analysis - will be necessary before it can be considered further.

Reviewer #1:

The authors investigate the temporal and spatial variability in gene expression between individual cells, in the context of a mammalian tissue. Specifically, the expression of prolactin in rat pituitary slices, both in the adult and during development. Pursuing questions of stochasticity and of cell-cell communication in the complex context of a multicellular tissue has the potential for significant insights, and some of the data presented by the authors is quite interesting, but I found the work deficient in terms of the quantitative and theoretical analysis (which the authors describe as "sophisticated", paragraph four, Introduction). The standard for such analysis is quite high, even for studies in mammalian tissues, as demonstrated e.g. by the recent work from Halpern et al. in mouse liver (Molecular Cell 2015; cited by the authors).

Specific points:

1) The authors analyze their temporal fluorescence data using a theoretical stochastic model, to extract the parameters of gene activity. However, the validity of the model is not challenged by testing that it can successfully reproduce the experimental data. Nowhere do we see a plot with direct comparison between an experimental observable and its predicted value from the model. Thus what we have is a "forward only" process where experimental data is fed into an algorithm, to produce estimated parameters, without the critical feedback from model to experiments. This dramatically diminishes the value of the theoretical analysis.

2) On a related note, it is unclear what was learned from the model that was not already in the experimental data. In particular, the temporal and spatial correlations in expression, and how those change during development, are evident from the straightforward analysis of the data, and it was not obvious that the model sheds any additional light on these findings.

3) Besides the theoretical model, the quantitative analysis of data was itself quite flawed in a number of instances. Examples:

i) "fluorescence activity showed a clear deviation from a white noise process indicating a pulsatile transcriptional behavior". Why would the expectation be of "white noise"? And why would deviation from white noise indicate pulses? This is very unclear.

ii) The cell-cell correlations in Figure 2-ii show an increase at large distances. This is probably an artifact, since it can be seen in the randomized control as well, but the presence of this trend is suspicious-could it indicate a flaw in the correlation calculation, e.g. in normalizing for the number of cell pairs?

Reviewer #2:

In this study by Featherstone and coworkers, the authors use single-cell imaging to look at prolactin expression in rat pituitary tissue. Expression is measured via a GFP reporter driven by the prolactin promoter in the context of a transgenic BAC. From these fluorescence time traces, the authors reconstruct transcriptional dynamics through a statistical inference model. Single-cell imaging of prolactin promoter activity in cells in culture and the inference model have been published previously by some of the authors. The novelty in this work lies in performing these measurements in a pituitary tissue slice. Although this approach has some caveats (inference of transcription dynamics from protein time-series, transgenic regulation, tissue slices, etc.), my opinion is that it is overall a reasonably faithful approximation of the actual endogenous regulation of this promoter in a tissue context. In that sense, it is a major technical advance. I am unaware of any other studies that look at stochastic gene regulation with this resolution in tissue. The major conclusions they reach are that the prolactin promoter is pulsatile in tissue, the pulsatility changes during development, and that cells show local coordination by means of adherens and gap junctions.

Overall, the data is of very high quality, and the analysis seems sound. What is missing, in my opinion, is biological insight into the functioning of this promoter in the context of the gland. However, I believe the authors have this understanding and may have even tried to convey it, but this referee didn't quite grasp the implications. It is possible that a revised manuscript with some restructuring and slightly re-directed analysis would make this message more appropriate for the broad readership of *eLife*.

1) My main criticism is on the nature of the developmental progression, gene output, and spatial coordination. The authors present a series of observations about all three, but I suggest that it would be more biologically insightful to integrate these observations to describe the overall output of prolactin from the tissue. Does overall output increase during developmental stages? If so, does this output result from more cells (i.e. greater density) or changes in the pulsing? If cells are pulsing in synchrony, would the overall output also display pulsatile production from the gland? It is tempting to conclude form the authors' data that cells have some sort of fixed prolactin regulatory circuit, but as the gland matures, more cells result in higher levels of output, and this output becomes dynamically synchronized through cell-cell communication. Is this correct? Or am I misunderstanding the data? Here, the quantitative modeling would be immensely valuable, perhaps allowing one to extrapolate from the tissue slices to the function in vivo.

[Editors' note: further revisions were requested prior to acceptance, as described below.]

Thank you for submitting your work entitled "Spatially Coordinated Dynamic Gene Transcription in Living Pituitary Tissue" for consideration by *eLife*. Your article has been reviewed by two peer reviewers, and the evaluation has been overseen by a Reviewing Editor and Jim Kadonaga as the Senior Editor.

The reviewers have discussed the reviews with one another and the Reviewing editor has drafted this decision to help you prepare a revised submission.

As you can see the reviewers are still unsure of your presentation of the significance of the results and their treatment in the manuscript. In their discussion they have stated that the work does not advance enough from previous published work. *eLife* requires that the manuscript reports on significantly new observations. Whether this manuscript is accepted will depend on how you can bring out the broader significance of your observations to their satisfaction.

Summary and essential revisions:

In this revision by Featherstone et al., the authors have made some textual and organizational changes to the manuscript to better convey the biological meaning. However, I still find the manuscript lacking in its appeal to a broad audience or even a narrower gene expression readership. It just doesn't go quite far enough in my opinion in generating new insight into the developing pituitary gland. Although I am strongly attracted to the approach the authors have developed, it seems to be an extension of their previous work in this area, published in a series of papers which observe the pulsatility, the change during development, and the refractory period (Harper, JCS, 2010; Featherstone, JCS, 2011; Harper, PLoS Bio, 2011). Here, they have refined their model of the underlying dynamics using a statistical inference method. They conclude that transcription dynamics do not obey a simple telegraph process but do not go beyond this phenomenology, nor do they dissect the functional consequences.

The real advance in this paper is the observed spatial coupling between individual cells and the finding that this spatial coupling changes over development. This coupling, which occurs over a length scale of ~ 30 μM, is perturbed by limited trypsin digestion. This finding is indeed an interesting one, but does it rise to the level of a self-sufficient story? And could it not be delivered in a more compelling way? In summary, I find the work to be technically accomplished, but my overall impression is that they have not spelled out clearly the significant advance in our biological understanding.

---

## [Author Response]

The Reviewers find the work of interest, in particular that the spatial relationship of cells within a tissue may lead to a common basis for expression. However they feel that there are some obscure analyses and the text in ambiguous in parts. One reviewer "found the work deficient in terms of the quantitative and theoretical analysis". The other thought that" the primary contribution of this paper may be the relationship between this stochastic process and the integrated production from the tissue" but that this was not clearly developed. In both cases, some extensive revisions - in particular, more rigorous text and analysis - will be necessary before it can be considered further.

We have made a series of detailed changes to the paper in response to these comments, and included a new final figure, a new supplemental figure, and revisions to original Figure 2 and Figure 3, to try to present the data more clearly and the key concepts more succinctly. We believe that these changes have significantly improved the manuscript, and hope that it will now be acceptable for publication in *eLife*. Our detailed responses are listed below.

Reviewer #1:

1) The authors analyze their temporal fluorescence data using a theoretical stochastic model, to extract the parameters of gene activity. However, the validity of the model is not challenged by testing that it can successfully reproduce the experimental data. Nowhere do we see a plot with direct comparison between an experimental observable and its predicted value from the model. Thus what we have is a "forward only" process where experimental data is fed into an algorithm, to produce estimated parameters, without the critical feedback from model to experiments. This dramatically diminishes the value of the theoretical analysis.

The reviewer cites the paper of Halpern et al. as an exemplar for this type of quantitative analysis. We agree that this paper represents the current cutting edge of the field and cite it in our manuscript. Halpern et al. employ sophisticated smFISH, which means that transcription dynamics are calculated from a single frozen snapshot of activity. In contrast, our approach tracks transcription dynamics in single cells in real time over 40h periods. The approach we have used is a significant development of work first published in two pioneering 2011 papers that used quantitative single cell time-course analysis (Suter et al. in Science, and our previous PLoS Biology paper by Harper et al.). The key novelty of the current work is the analysis of transcriptional bursting kinetics in relation to spatial proximity of cells within intact living tissue. This uses major advances in the statistical analysis of multi-state transcription, based on development and application of the theoretical tools described in our Biostatistics paper (Hey et al. 2015).

The reviewer may not have appreciated that the development of our statistical analysis fundamentally relied on iterative testing of the model against synthetic and real data. Our approach is to date the only way of comparing the modelling to the true underlying biological process (assuming that the inferred switches really are switches in transcription). We confirmed that the model reasonably fits the experimental data, iteratively tested through recursive residuals. These residuals were calculated on both the P1.5 and Adult tissue that were presented in the Biostatistics paper and we are including them for your consideration. These residuals are a way of comparing the prediction from the model (based on the estimates) and the observed data. Detailed information about the use of such residuals is contained in Appendix G of the Biostatistics paper (see supplementary material available at Biostatistics online) and no departure from the model assumptions was detected, indicating that the SSM under both the LNA and BDA fits the data well.

We have added the following sentence to the manuscript referring to these points: “The fit of the model was tested through calculation of recursive residuals as a way of comparing the prediction from the model and the observed data (for information see Appendix G of (20)). These showed no departure from the model assumptions, indicating that the SSM fitted the data well.”

2) On a related note, it is unclear what was learned from the model that was not already in the experimental data. In particular, the temporal and spatial correlations in expression, and how those change during development, are evident from the straightforward analysis of the data, and it was not obvious that the model sheds any additional light on these findings.

This is a critical point. The experimental data referred to by the reviewer is raw fluorescent protein expression data. The fluorescence signal depends on multiple processes including transcription, translation, and protein and mRNA degradation. In order to probe the correlation between cells, it is critical to be able to accurately measure, and statistically assess, multi-state switches in transcription rate in isolation from other processes. Back-calculation of transcription rate, as we have undertaken, allows us to estimate the correlation between cells in terms of the transcriptional process by assessing correlation in the switch times and switch directions. The modelling approach also revealed the presence of shorter “on” durations in immature pituitary tissue and that similar distributions of transcription rate were present in all stages of pituitary development. These insights are important for understanding how changes in individual cell responses lead to changes to the overall output from a tissue system. The importance of modelling transcriptional dynamics and the insights gained from such analyses, including the modelling approach undertaken in our study were discussed in paragraph two, Discussion. In terms of informing conclusions regarding the overall biological behaviour we have added some further text to the Discussion in paragraph six and now include a new simulation model and hypothesis regarding the nature of overall transcriptional output from an endocrine tissue (outlined in Figure 9).

3) Besides the theoretical model, the quantitative analysis of data was itself quite flawed in a number of instances. Examples:

*i) "fluorescence activity showed a clear deviation from a white noise process indicating a pulsatile transcriptional behavior". Why would the expectation be of "white noise"? And why would deviation from white noise indicate pulses? This is very unclear.*

*ii) The cell-cell correlations in Figure 2-ii show an increase at large distances. This is probably an artifact, since it can be seen in the randomized control as well, but the presence of this trend is suspicious-could it indicate a flaw in the correlation calculation, e.g. in normalizing for the number of cell pairs?*

Example i). The reviewer is correct to criticise the sentence “Autocorrelation analyses did not identify a regular homogeneous period, but fluorescence activity showed a clear deviation from a white noise process indicating a pulsatile transcriptional behaviour (Figure 1) with a definite quantifiable statistical structure. “We agree that this was misleading since it seems to suggest that since the process was not a white noise process it must therefore represent pulsatile transcriptional behaviour. What we should have said was:

“Autocorrelation analyses did not identify a regular homogeneous period, and fluorescence activity showed a clear deviation from a white noise process (Figure 1). Further, analysis using a Stochastic Switch Model (SSM) uncovered a definite quantifiable statistical structure indicating a pulsatile transcriptional behaviour compatible with the pulsed telegraph process as described above.”

This sentence has been modified accordingly in the manuscript in subheading “Patterns of Prolactin Gene Transcription Activity in Adult Pituitary Tissue”.

Example ii) The referee picks up an important point that we have tested further. The cell-cell correlations in Figure 2 that the referee refers to are inside the 90% confidence interval of the randomised control and therefore are not significant. This is further emphasised by the data in part i) of the figure from which it is clear that the number of pairs in the bins above 70 are very small. We have addressed this by re-plotting the data using bins chosen so that they all have the same number of cell pairs in them. This clearly shows the significant decrease in correlation for distances less than about 40μm and we see that the last two bins fall firmly in the 90% confidence interval of the randomised control. This new information is now included in Figure 3—figure supplement 1.

Reviewer #2:

1) My main criticism is on the nature of the developmental progression, gene output, and spatial coordination. The authors present a series of observations about all three, but I suggest that it would be more biologically insightful to integrate these observations to describe the overall output of prolactin from the tissue. Does overall output increase during developmental stages? If so, does this output result from more cells (i.e. greater density) or changes in the pulsing? If cells are pulsing in synchrony, would the overall output also display pulsatile production from the gland? It is tempting to conclude form the authors' data that cells have some sort of fixed prolactin regulatory circuit, but as the gland matures, more cells result in higher levels of output, and this output becomes dynamically synchronized through cell-cell communication. Is this correct? Or am I misunderstanding the data? Here, the quantitative modeling would be immensely valuable, perhaps allowing one to extrapolate from the tissue slices to the function in vivo.

We agree that we should propose the biological significance and context of the work more clearly. We have tried to address this point by drawing together the various strands in new paragraphs in the Discussion section, paragraph six. We hope these significantly improve the paper and make clearer the in vivo relevance of the multi-state transcription model that we are proposing. We also include a schematic diagram together with new data from a simulation model to help respond to this point (Figure 9).

[Editors' note: further revisions were requested prior to acceptance, as described below.]

In this revision by Featherstone et al., the authors have made some textual and organizational changes to the manuscript to better convey the biological meaning. However, I still find the manuscript lacking in its appeal to a broad audience or even a narrower gene expression readership. It just doesn't go quite far enough in my opinion in generating new insight into the developing pituitary gland. Although I am strongly attracted to the approach the authors have developed, it seems to be an extension of their previous work in this area, published in a series of papers which observe the pulsatility, the change during development, and the refractory period (Harper, JCS, 2010; Featherstone, JCS, 2011; Harper, PLoS Bio, 2011). Here, they have refined their model of the underlying dynamics using a statistical inference method. They conclude that transcription dynamics do not obey a simple telegraph process but do not go beyond this phenomenology, nor do they dissect the functional consequences.

The real advance in this paper is the observed spatial coupling between individual cells and the finding that this spatial coupling changes over development. This coupling, which occurs over a length scale of ~ 30 μM, is perturbed by limited trypsin digestion. This finding is indeed an interesting one, but does it rise to the level of a self-sufficient story? And could it not be delivered in a more compelling way? In summary, find the work to be technically accomplished, but my overall impression is that they have not spelled out clearly the significant advance in our biological understanding.

One reviewer questioned the significance of our study for a broad biological audience. We believe that this paper not only presents important advances in both theoretical and experimental approach, but is also the first study with sufficient resolution and throughput to assess the effects of tissue environment on spatial coordination of gene expression. Overall, the paper represents a complete integrated and quantitative analysis of the control of transcription dynamics, both temporal and spatial, across development in a key mammalian tissue. No previous study has achieved this.

The main changes we have made are as follows:

We have re-written the Abstract to make the main messages of the paper clearer.

We have simplified and streamlined the Introduction in order to present the main background questions more clearly. In response to a referee’s minor comments we have also removed subjective comments about significance or importance.

We have not altered the figures beyond minor adjustments to labelling, but we have tried to clarify the description of the results shown in the figures in order to make the key messages more accessible to a broad readership. Changes in the Results section particularly concern the treatment of the results shown in Figure 2, Figure 4 and Figure 7, where we have tried to clarify the presentation in the main text. We have also revised the figure legends slightly (especially Figure 7), and adjusted the figure labeling (Figure 2, Figure 5, Figure 7, Figure 8), for greater clarity.

Changes in the Discussion were made in the last two paragraphs, to emphasise the key new findings, which we have tried to summarise more succinctly. These findings are:

1) Multi-state modelling of transcription dynamics indicates that transcription pulses are not simply binary in nature, and we characterise switching behaviour in intact tissue.

2) Both the duration of pulses and the coordination of transcriptional dynamics differ between immature and adult pituitary tissue.

3) Tissue structure affects cellular behaviour, and exerts local coordination of transcription dynamics. Our work with not only trypsin, but also gap junction blockers, indicates that gap junction signalling is involved in this behaviour.